# Bridge-IF: Learning Inverse Protein Folding with Markov Bridges

**Yiheng Zhu**[1], **Jialu Wu**[2], **Qiuyi Li**[3], **Jiahuan Yan**[1], **Mingze Yin**[4], **Wei Wu**[5],
**Mingyang Li**[3], **Jieping Ye**[3], **Zheng Wang**[3]*, **Jian Wu**[1,4,6]*

[1]College of Computer Science & Technology and Liangzhu Laboratory, Zhejiang University
[2]College of Pharmaceutical Sciences, Zhejiang University
[3]Alibaba Cloud Computing
[4]School of Public Health, Zhejiang University
[5]School of Artificial Intelligence and Data Science, University of Science and Technology of China
[6]The Second Affiliated Hospital Zhejiang University School of Medicine
{zhuyiheng2020, jialuwu, jyansir, yinmingze, wujian2000}@zju.edu.cn
{liqiuyi.lqy, sangheng.lmy, yejieping.ye, wz388779}@alibaba-inc.com
urara@mail.ustc.edu.cn

## Abstract

Inverse protein folding is a fundamental task in computational protein design, which aims to design protein sequences that fold into the desired backbone structures. While the development of machine learning algorithms for this task has seen significant success, the prevailing approaches, which predominantly employ a discriminative formulation, frequently encounter the error accumulation issue and often fail to capture the extensive variety of plausible sequences. To fill these gaps, we propose Bridge-IF, a generative diffusion bridge model for inverse folding, which is designed to learn the probabilistic dependency between the distributions of backbone structures and protein sequences. Specifically, we harness an expressive structure encoder to propose a discrete, informative prior derived from structures, and establish a Markov bridge to connect this prior with native sequences. During the inference stage, Bridge-IF progressively refines the prior sequence, culminating in a more plausible design. Moreover, we introduce a reparameterization perspective on Markov bridge models, from which we derive a simplified loss function that facilitates more effective training. We also modulate protein language models (PLMs) with structural conditions to precisely approximate the Markov bridge process, thereby significantly enhancing generation performance while maintaining parameter-efficient training. Extensive experiments on well-established benchmarks demonstrate that Bridge-IF predominantly surpasses existing baselines in sequence recovery and excels in the design of plausible proteins with high foldability. The code is available at `https://github.com/violet-sto/Bridge-IF`.

## 1 Introduction

Proteins are 3D folded linear chains of amino acids that execute the myriad of biological processes fundamental to life, such as catalysing metabolic reactions, mediating immune responses, and responding to stimuli [23]. Designing protein sequences that fold into desired 3D structures, known as inverse protein folding, is a crucial task with great potential for applications in protein engineering [29, 66, 5]. Beyond long-established physics-based methods like Rosetta [2], the considerable promise of leveraging geometric deep learning for protein structure modeling has given rise to an ongoing

---

*Corresponding authors.

38th Conference on Neural Information Processing Systems (NeurIPS 2024).

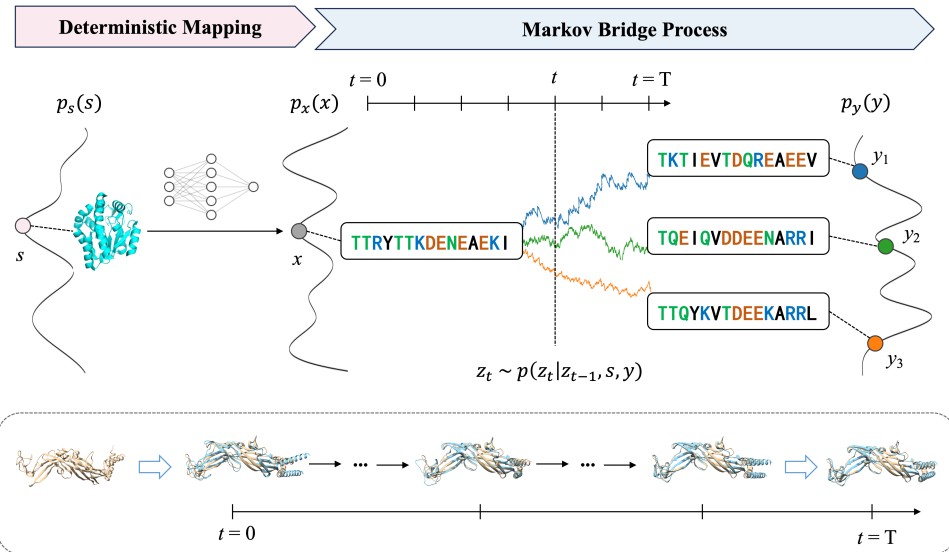

Figure 1: **Overview of Bridge-IF**. Bridge-IF consists of an expressive structure encoder supervised by native sequences for proposing a discrete, deterministic prior, and a Markov bridge model for learning the dependency between the distribution of prior sequences and the distribution of native sequences. During the inference stage, Bridge-IF progressively refines the prior sequence.

paradigm. This paradigm is centered on deciphering the principles of protein design directly from data and on predicting sequences corresponding to specific structures [25, 27, 6, 22].

Despite substantial advancements, most existing approaches follow a discriminative formulation for learning inverse folding [58], consequently encountering two principal obstacles: (i) *Error accumulation issue*. For instance, Transformer-based autoregressive models are constrained by their inherent sequential generation process and exposure bias, which prevents them from correcting preceding erroneous predictions. (ii) *One-to-many mapping nature of the inverse folding problem.* A multitude of distinct amino acid sequences possess the capability to fold into an identical protein backbone structure, a phenomenon exemplified by homologous proteins. Discriminative models are incapable of capturing the one-to-many mapping from the protein structure to non-unique sequences, thereby facing difficulties in covering the broad spectrum of plausible solutions [58].

Recent studies have advanced the iterative refinement strategy to optimize the previously generated results, aiming to reduce prediction errors [64, 14, 42]. These approaches employ a refinement module to identify and correct inaccurately predicted amino acids. However, as the number of refinement iterations grows, managing the intermediate stages effectively becomes more challenging, potentially hindering sustained performance gains.

Diffusion-based generative models [45, 17], particularly their discrete extensions [3], which offer a structured iterative refinement process with probabilistic interpretation, appear to be a promising solution. GraDe-IF [58] is a pioneer in investigating diffusion models for inverse folding, leveraging the backbone structure to guide the denoising process on the amino acid residues. However, as diffusion models are designed to learn a single intractable data distribution, the prior distribution utilized by GraDe-IF is restricted to a simple noise distribution (i.e., a uniform distribution across all residue types), which has little or no information about the distribution of native sequences. It remains unclear whether this default formulation best suits conditional generative problems such as inverse protein folding, where the backbone structures provide significantly more information than random noise. Thus, an exciting research question naturally arises: *Can we propose a more strong and informative prior based on desired backbone structures to enhance the quality of samples and accelerate the inference process?*

In this work, we propose Bridge-IF, a novel generative diffusion bridge model for inverse folding. Its core design is aimed at generating protein sequences from a structure-aware prior. As shown

in Figure 1, we leverage an expressive structure encoder supervised by native sequences to propose a discrete, deterministic prior based on desired structures, and build a Markov bridge [10, 24] between it and the native sequence. By approximating the reference Markov bridge process, Bridge-IF learns to progressively refine the prior sequence, resulting in a more plausible design. Furthermore, we present a fresh reparameterization perspective on Markov bridge models and derive a simplified loss function that yields enhanced training effectiveness. Inspired by significant advances in protein language models (PLMs) for understanding proteins [9, 34], we innovatively integrate conditions, including timestep and structures, into PLMs to accurately approximate the Markov bridge process. This approach notably improves generation performance while ensuring parameter-efficient training. Empirically, we demonstrate that Bridge-IF outperforms state-of-the-art baselines on several standard benchmarks and excels in the design of plausible proteins with high foldability.

To summarise, the main contributions of this work are as follows:

- We introduce Bridge-IF, the first generative diffusion bridge model based on Markov bridges for inverse folding. We also offer a reparameterization perspective and derive a simplified loss function to facilitate effective training.

- We innovatively adapt PLMs to effectively capture both timestep and structural information while ensuring the modified architecture is compatible with pre-trained weights.

- Experiments verify that Bridge-IF achieves state-of-the-art performance on standard benchmarks.

## 2   Related work

### 2.1   Inverse protein folding

Recently, AI algorithms have spurred a revolution in modeling protein folding [28, 34]. Meanwhile, the inverse problem of protein folding, which aims to infer an amino acid sequence that will fold into the desired structure, is gaining increasing attention [6]. By representing protein backbone structures as a $k$-NN graph, geometric deep learning has achieved remarkable progress in learning inverse folding [25, 6, 22], surpassing traditional physics-based approaches [2], and even facilitating the design of a range of experimentally validated proteins [6, 56]. Modern deep learning-based inverse folding approaches typically comprise a structure encoder and a sequence decoder. Depending on their decoding strategies, these approaches can be classified into three categories: autoregressive models, one-shot models, and iterative models. Most methods adopt the autoregressive decoding scheme to generate amino acid sequences [25, 6, 22]. Given that autoregressive models tend to have low inference speed, some researchers have investigated one-shot methods that facilitate the parallel generation of multiple tokens [12, 38]. Since directly predicting highly plausible sequences is challenging, some works have shifted their attention to iterative refinement [64, 14, 26, 42, 58]. For instance, LM-Design [64] and KW-design [14] utilize the pre-trained knowledge from PLMs to reconstruct a native sequence from a corrupted version. The Potts model-based ChromaDesign [26] and CarbonDesign [42] employ iterative sampling techniques, including Markov chain Monte Carlo, to design protein sequences. GraDe-IF [58] further leverages the principles of discrete denoising diffusion probabilistic models [3], demonstrating a strong capacity to encompass diverse plausible solutions. In this work, we present the first generative diffusion bridge model for inverse folding.

### 2.2   Diffusion models

Diffusion-based generative models [45, 17] have showcased remarkable successes in a wide range of applications, ranging from image synthesis [8], audio synthesis [31], to video generation [18]. Generally, the essential idea behind these models is to define a forward diffusion process that gradually transforms the data into a simple prior distribution and learn a reverse denoising process to gradually recover original data samples from the prior distribution. While most existing methods are designed for modeling continuous data, a few efforts have extended diffusion models to discrete data domains [3, 33, 52, 36]. Recently, diffusion models have also found utility in scientific discovery [55], particularly in protein design [56, 59, 1, 15, 58].

## 2.3 Schrödinger bridge problem

The Schrödinger bridge (SB) problem is a classical entropy-regularized optimal transport problem [43, 32, 4]. Given a data distribution, a prior distribution, and a reference stochastic process between them, solving the SB problem amounts to finding the closest process to the reference in terms of Kullback-Leibler divergence on path spaces. This concept exhibits fundamental similarities to diffusion models [47], particularly in the field of unconditional generative modeling [49, 54, 7, 44], where the prior distribution assumes the form of Gaussian noise. Notably, SB formalism offers a general framework for approximating the reference stochastic process by training on coupled samples from two continuous distributions [19, 46, 35, 65]. The recently proposed Markov bridge [10, 24] has broadened the scope of the SB, enabling it to model categorical distributions. In this work, we present the first diffusion bridge model for inverse protein folding.

## 3 Background

### 3.1 Problem formulation and notation

Generally, a protein can be represented as a pair of amino acid sequence and structure $(\boldsymbol{y}, \boldsymbol{s})$, where $\boldsymbol{y} = [y_1, y_2, \ldots, y_n]$ denotes its sequence of $n$ residues with $y_i \in \{1, 2, \ldots, 20\}$ indicating the type of the $i$-th residue, and $\boldsymbol{s} = [s_1, s_2, \ldots, s_n] \in \mathbb{R}^{n \times 4 \times 3}$ denotes its structure with $s_i$ representing the Cartesian coordinates of the $i$-th residue's backbone atoms (i.e., N, C-$\alpha$, and C, with O optionally). The inverse protein folding problem aims to automatically identify the protein sequence $\boldsymbol{y}$ that can fold into the given structure $\boldsymbol{s}$. Given that homologous proteins invariably exhibit similar structures, the solution for a given structure is not unique [16]. Hence, an ideal model, parameterized by $\theta$, should be capable of learning the underlying mapping from protein backbone structures to their corresponding sequence distributions $p_\theta(\boldsymbol{y}|\boldsymbol{s})$.

### 3.2 Markov bridge models

Markov bridge model [24] is a general framework for learning the probabilistic dependency between two intractable discrete-valued distributions $p_\mathcal{X}$ and $p_\mathcal{Y}$. For a pair of samples $(\boldsymbol{x}, \boldsymbol{y}) \sim p_{\mathcal{X}, \mathcal{Y}}(\boldsymbol{x}, \boldsymbol{y})$, it defines a Markov process pinned to fixed start and end points $\boldsymbol{z}_0 = \boldsymbol{x}$ and $\boldsymbol{z}_T = \boldsymbol{y}$ through a sequence of random variables $(\boldsymbol{z}_t)_{t=0}^T$ that satisfies the Markov property,

$$p(\boldsymbol{z}_t|\boldsymbol{z}_0, \boldsymbol{z}_1, \ldots, \boldsymbol{z}_{t-1}, \boldsymbol{y}) = p(\boldsymbol{z}_t|\boldsymbol{z}_{t-1}, \boldsymbol{y}). \tag{1}$$

To pin the process at the end point $\boldsymbol{z}_T = \boldsymbol{y}$, we have an additional requirement,

$$p(\boldsymbol{z}_T = \boldsymbol{y}|\boldsymbol{z}_{T-1}, \boldsymbol{y}) = 1. \tag{2}$$

Assuming that both $p_\mathcal{X}$ and $p_\mathcal{Y}$ are categorical distributions with a finite sample space $\{1, \ldots, K\}$, we can represent data points as one-hot vectors: $\boldsymbol{x}, \boldsymbol{y}, \boldsymbol{z}_t \in \{0, 1\}^K$, and define the transition probabilities (Equation 1) as follows,

$$p(\boldsymbol{z}_{t+1}|\boldsymbol{z}_t, \boldsymbol{y}) = \text{Cat}\left(\boldsymbol{z}_{t+1}; \boldsymbol{Q}_t \boldsymbol{z}_t\right), \tag{3}$$

where $\text{Cat}(\cdot \, ; \boldsymbol{p})$ is a categorical distribution with probabilities given by $\boldsymbol{p}$, and $\boldsymbol{Q}_t$ is a transition matrix parameterized as

$$\boldsymbol{Q}_t := \boldsymbol{Q}_t(\boldsymbol{y}) = \beta_t \boldsymbol{I}_K + (1 - \beta_t)\boldsymbol{y}\mathbf{1}_K^\top, \tag{4}$$

where $\beta_t$ is a schedule parameter transitioning from $\beta_0 = 1$ to $\beta_{T-1} = 0$. It is easy to see that $\boldsymbol{z}_t$ can be efficiently sampled from $p(\boldsymbol{z}_{t+1}|\boldsymbol{z}_0, \boldsymbol{z}_T) = \text{Cat}\left(\boldsymbol{z}_{t+1}; \overline{\boldsymbol{Q}}_t \boldsymbol{z}_0\right)$ with a cumulative product matrix $\overline{\boldsymbol{Q}}_t = \boldsymbol{Q}_t \boldsymbol{Q}_{t-1}...\boldsymbol{Q}_0 = \overline{\beta}_t \boldsymbol{I}_K + (1 - \overline{\beta}_t)\boldsymbol{y}\mathbf{1}_K^\top$, where $\overline{\beta}_t = \prod_{s=0}^t \beta_s$.

**Training** Using the finite set of coupled samples $\{(\boldsymbol{x}_i, \boldsymbol{y}_i)\}_{i=1}^D \sim p_{\mathcal{X}, \mathcal{Y}}$, Markov bridge model learns to sample $\boldsymbol{y}$ when only $\boldsymbol{x}$ is available by approximating $\boldsymbol{y}$ with a neural network $\varphi_\theta$:

$$\hat{\boldsymbol{y}} = \varphi_\theta(\boldsymbol{z}_t, t), \tag{5}$$

and defining an approximated transition kernel,

$$q_\theta(\boldsymbol{z}_{t+1}|\boldsymbol{z}_t) = \text{Cat}\left(\boldsymbol{z}_{t+1}; \boldsymbol{Q}_t(\hat{\boldsymbol{y}})\boldsymbol{z}_t\right). \tag{6}$$

$\varphi_\theta$ is trained by optimizing the variational bound on negative log-likelihood $\log q_\theta(\boldsymbol{y}|\boldsymbol{x})$, which has the following closed-form expression,

$$-\log q_\theta(\boldsymbol{y}|\boldsymbol{x}) \leq T \cdot \mathbb{E}_{t \sim \mathcal{U}(0,\ldots,T-1)} \underbrace{\mathbb{E}_{\boldsymbol{z}_t \sim p(\boldsymbol{z}_t|\boldsymbol{x}, \boldsymbol{y})} D_{\text{KL}}\left(p(\boldsymbol{z}_{t+1}|\boldsymbol{z}_t, \boldsymbol{y}) \| q_\theta(\boldsymbol{z}_{t+1}|\boldsymbol{z}_t)\right)}_{\mathcal{L}_t}. \tag{7}$$

**Sampling** To sample a data point $\boldsymbol{y} \equiv \boldsymbol{z}_T$ starting from a given $\boldsymbol{z}_0 \equiv \boldsymbol{x} \sim p_{\mathcal{X}}(\boldsymbol{x})$, one can iteratively predict $\hat{\boldsymbol{y}} = \varphi_\theta(\boldsymbol{z}_t, t)$ and then derive $\boldsymbol{z}_{t+1} \sim q_\theta(\boldsymbol{z}_{t+1}|\boldsymbol{z}_t) = \text{Cat}(\boldsymbol{z}_{t+1}; \boldsymbol{Q}_t(\hat{\boldsymbol{y}})\boldsymbol{z}_t)$ for $t = 0, \ldots, T - 1$.

## 4 Methods

In this section, we introduce Bridge-IF, a Markov bridge-based model for inverse protein folding. Figure 1 shows an overview of our proposed Bridge-IF. Due to space limitation, we present the detailed algorithm in Appendix A. To begin, we describe how to extend Markov bridge techniques to facilitate the inverse protein folding task. Next, we propose a simplified training objective. Finally, we elucidate how to modulate pre-trained PLMs with structural conditions to approximate the Markov bridge process.

### 4.1 Overview of Bridge-IF

We frame the inverse protein folding problem as a generative problem of modeling a stochastic process between the distributions of backbone structures $p_{\mathcal{S}}(s)$ and protein sequences $p_{\mathcal{Y}}(y)$. As previously discussed, diffusion bridge models, with their general properties of an unrestricted prior form, serves as an ideal substitution for diffusion models in the presence of a well-defined informative prior. Regrettably, to the best of our knowledge, no existing method can directly model the dependency between two distinct types of distributions: specifically, the *continuous* source distribution of backbone structures and the *discrete* target distribution of protein sequences.

To reconcile the differences between source and target distributions and streamline the modeling process, we propose introducing a discrete proposal distribution to serve as a deterministic prior. We parameterize the proposal distribution using a structure encoder $\mathcal{E} : \mathcal{S} \rightarrow \mathcal{X}$ that is supervised by ground-truth target sequences. Recent advancements have demonstrated that an expressive encoder is capable of directly predicting pretty good protein sequences in a one-shot manner [12]. This approach enables us to utilize structural information more effectively, rather than simply employing it to guide the denoising process as in previous diffusion-based methods like GraDe-IF [58]. In this work, we will take the discriminative model PiFold [12] as the structure encoder to produce a clean and deterministic prior $\boldsymbol{x} = \mathcal{E}(\boldsymbol{s})$. Upon this deterministic mapping from structure to sequence, we simplify the originally complex problem of modeling $p(\boldsymbol{s}, \boldsymbol{y})$ into the more tractable problem of modeling $p(\boldsymbol{x}, \boldsymbol{y})$. Then, we build a Markov bridge [10, 24] between the prior sequence and the native sequence to model the stochastic process, leading to a data-to-data process. As depicted in the lower half of Figure 1, each sampling step progressively refines the prior sequence, which contains significant information about the target sequence, ultimately resulting in a more precise prediction.

Recall that the Markov bridge models are typically trained by optimizing the variational bound on negative log-likelihood $\log q_\theta(\boldsymbol{y}|\boldsymbol{x})$ (Equation 7), which is analytically complicated and hard to optimize in practice [63, 62]. Therefore, we here propose a reparameterization perspective on Markov bridge models, deriving a simplified loss function for easier optimization (§4.2).

We build Markov bridges in the sequence space, treating the sequence representation as a set of independent categorical random variables. To model the Markov bridge process, $\boldsymbol{Q}_t$ is applied separately to each residue within a protein sequence. Motivated by the impressive advancements in PLMs for understanding and generating proteins [9, 34, 37, 60], we advocate for employing PLMs to approximate the Markov bridge process. This approach capitalizes on the emergent evolutionary knowledge of proteins, learned from an extensive dataset of protein sequences. Additionally, we utilize the latent structural features extracted by the structure encoder to prompt PLMs, thereby guiding the generation of structurally coherent proteins. Formally, the final state of the Markov bridge process is approximated by $\hat{\boldsymbol{y}} = \varphi_\theta(\boldsymbol{z}_t, \boldsymbol{s}, t)$, foregoing the use of Equation 5. We investigate the integration of conditional information, such as timestep and structure, into PLMs, focusing on preserving their emergent knowledge and achieving parameter-efficient training (§4.3).

### 4.2 Reparameterized Markov bridge models

Inspired by the similarities between Markov bridge models [24] and discrete diffusion models [3, 63], we propose a reparameterization of the Markov bridge model characterized in §3.2 to enable more effective training. With the reparameterization trick, we introduce a latent binary random variable

$v_t \sim \text{Bernoulli}(\overline{\beta}_{t-1})$ to indicate whether $\boldsymbol{z}_t$ has been transformed from $\boldsymbol{z}_0$ to $\boldsymbol{z}_T$. Thus $\boldsymbol{z}_t$ can be sampled from $p(\boldsymbol{z}_t|v_t, \boldsymbol{z}_0, \boldsymbol{y}) = v_t\boldsymbol{z}_0 + (1 - v_t)\boldsymbol{y}$. Accordingly, $p(\boldsymbol{z}_{t+1}|\boldsymbol{z}_t, \boldsymbol{y})$ can be equivalently written as:

$$p(\boldsymbol{z}_{t+1}|v_t, \boldsymbol{z}_t, \boldsymbol{y}) = \begin{cases} \boldsymbol{z}_t & \text{if } v_t = 0 \\ (1 - \beta_t)\boldsymbol{y} + \beta_t\boldsymbol{z}_t & \text{if } v_t = 1 \end{cases} \tag{8}$$

Using the teacher-forcing approach, we can similarly define the approximation process as

$$q_\theta(\boldsymbol{z}_{t+1}|v_t, \boldsymbol{z}_t) = \begin{cases} \boldsymbol{z}_t & \text{if } v_t = 0 \\ (1 - \beta_t)\varphi_\theta(\boldsymbol{z}_t, t) + \beta_t\boldsymbol{z}_t & \text{if } v_t = 1 \end{cases} \tag{9}$$

**Proposition 4.1.** *The loss objective $\mathcal{L}_t(\theta)$ for sequence x at the t-th step can be reduced to the form*

$$\mathcal{L}_t(\theta) = \lambda_t \mathbb{E}_{p(\boldsymbol{z}_t|\boldsymbol{x}, \boldsymbol{y})}[-v_t\boldsymbol{y}^T \log \varphi_\theta(\boldsymbol{z}_t, t)], \tag{10}$$

*where $\lambda_t = 1 - \beta_t$.*

The full derivation is provided in C. This derived expression of $\mathcal{L}_t(\theta)$ formulates the training loss as a re-weighted standard multi-class cross-entropy loss function, which is computed over tokens that have not been transformed to the ground truth $\boldsymbol{y} = \boldsymbol{z}_T$. Following Ho et al. [17], we set $\lambda_t$ to a constant 1 in practice. Compared to the simpler cross-entropy loss calculated across all tokens, this new formulation places greater weight on tokens that require refinement. On the other hand, it is conceptually simpler than the original training loss (Equation 7), which requires calculating the complicated KL divergence between two categorical distributions $D_{\text{KL}}[p(\boldsymbol{z}_{t+1}|\boldsymbol{z}_t, \boldsymbol{y}) \| q_\theta(\boldsymbol{z}_{t+1}|\boldsymbol{z}_t)]$.

### 4.3   Network architecture design space

We adopt pre-trained PLMs as the base network to approximate the final state of the Markov bridge process. Typically, PLMs exclusively take protein sequences as input during the pre-training stage, making it non-trivial to integrate timestep and structural conditions into the PLMs. Hence, we innovatively tailor the Transformer blocks [50] to effectively capture timestep and structural information, as depicted in Figure 2. To facilitate efficient training, the architecture of our model is delicately designed for compatibility with the pre-trained weights. Our exposition emphasizes fundamental principles and the corresponding modifications to the base network.

#### 4.3.1   AdaLN-Bias

Inspired by DiT [41], we explore replacing standard layer norm layers in transformer blocks with adaptive layer norm (adaLN) to modulate the normalization's output based on both the timestep of the Markov bridge process and the backbone structure. The key idea is to regress the dimension-wise scale and shift parameters $\gamma$ and $\beta$ of the layer norm from the sum of the timestep embedding and the pooled structure representation. In our situation, meaningful pre-trained parameters $\gamma$ and $\beta$ are readily accessible. Upon commencing the fine-tuning stage, it is crucial that these parameters are close to the pre-trained values to preserve the effectiveness of the original model, since a poor initialization could significantly deteriorate performance. For simplicity, we propose to predict bias $\Delta\gamma$ and $\Delta\beta$ on the frozen original scalars and initialize the multi-layer perception (MLP) to output the zero-vector for all $\Delta\gamma$ and $\Delta\beta$. We term the proposed variant of adaLN as adaLN-Bias.

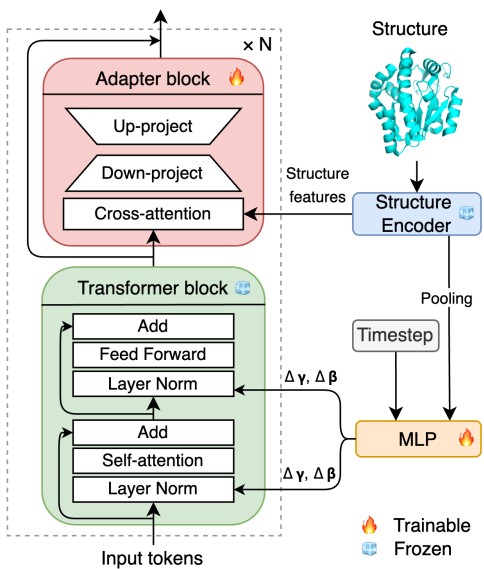

Figure 2: Model architecture of Bridge-IF.

Table 1: Results comparison on the **CATH** dataset. Benchmarked results are quoted from Hsu et al. [22], Zheng et al. [64], Yi et al. [58], Gao et al. [14]. †: "Single-chain" in Hsu et al. [22] is defined differently. The **best** and underline suboptimal results are labeled with bold and underline.

| | Model | Perplexity ↓ | | | Recovery Rate % ↑ | | |
|---|---|---|---|---|---|---|---|
| | | Short | Single-chain | All | Short | Single-chain | All |
| CATH v4.2 | StructGNN [25] | 8.29 | 8.74 | 6.40 | 29.44 | 28.26 | 35.91 |
| | GraphTrans [25] | 8.39 | 8.83 | 6.63 | 28.14 | 28.46 | 35.82 |
| | GCA [48] | 7.09 | 7.49 | 6.05 | 32.62 | 31.10 | 37.64 |
| | GVP [27] | 7.23 | 7.84 | 5.36 | 30.60 | 28.95 | 39.47 |
| | AlphaDesign [11] | 7.32 | 7.63 | 6.30 | 34.16 | 32.66 | 41.31 |
| | ProteinMPNN [6] | 6.21 | 6.68 | 4.61 | 36.35 | 34.43 | 45.96 |
| | PiFold [12] | 6.04 | 6.31 | 4.55 | 39.84 | 38.53 | 51.66 |
| | GraDe-IF [58] | **5.49** | 6.21 | 4.35 | **45.27** | 42.77 | 52.21 |
| | *With PLMs* | | | | | | |
| | LM-Design (ESM-1b 650M) [64] | 6.77 | 6.46 | 4.52 | 37.88 | 42.47 | 55.65 |
| | KW-Design (ESM-2 650M) [14] | 6.05 | 5.29 | 3.90 | 43.32 | 46.30 | 57.38 |
| | **Bridge-IF** (ESM-1b 650M) | 5.67 | **5.27** | **3.90** | 43.84 | **48.24** | **58.49** |
| | **Bridge-IF** (ESM-2 650M) | 5.68 | **5.06** | **3.83** | 43.86 | **48.96** | **58.59** |
| CATH v4.3 | GVP-large [22] | 7.68 | 6.12† | 6.17 | 32.60 | 39.40† | 39.20 |
| | ESM-IF [22] | 8.18 | 6.33† | 6.44 | 31.30 | 38.50† | 38.30 |
| | +1.2M AF2 predicted data [22] | 6.05 | 4.00† | 4.01 | 38.10 | 51.50† | 51.60 |
| | *With PLMs* | | | | | | |
| | LM-Design (ESM-1b 650M) [64] | 5.66 | 5.52 | 4.01 | 46.84 | 48.63 | 56.63 |
| | **Bridge-IF** (ESM-1b 650M) | **5.17** | **4.63** | **3.68** | **50.00** | **53.49** | **58.93** |

#### 4.3.2 Structural adapter

Considering that the pooled structure representation might only retain coarse-grained information, the network could consequently lack a detailed understanding of the structure input and necessitate information derived from original structural features to compensate. We incorporate a multi-head cross-attention module to the transformer block, enabling the network to flexibly interact with the structural features extracted from the structure encoder [64]. To facilitate pre-trained weights, we further integrate it into a bottleneck adapter layer [21] with residual connection, preserving the input for the subsequent layers.

*We stress that we freeze all pre-trained parameters of the base network during training.*

## 5 Experiments

In this section, we first demonstrate the effectiveness of our Bridge-IF on the standard CATH benchmark [40]. Next, we assess Bridge-IF for its applicability in *de novo* protein design. Moreover, we conduct several ablation studies to empirically justify the key design choices. Further results pertaining to the design of multi-chain protein complexes can be found in Appendix B.1.

### 5.1 Experimental protocol

**Training setup**  We conduct experiments on both **CATH v4.2** and **CATH v4.3**, where proteins are categorized based on the CATH hierarchical classification of protein structure, to ensure a comprehensive analysis. Following the standard data splitting provided by Ingraham et al. [25], CATH v4.2 dataset consists of 18,024 proteins for training, 608 proteins for validation, and 1,120 proteins for testing. Following the standard data splitting provided by Hsu et al. [22], CATH v4.3 dataset consists of 16,153 proteins for training, 1,457 proteins for validation, and 1,797 proteins for testing. For a fair comparison with iterative models [64, 14], we use pre-trained PiFold [12] to propose the prior distribution. We use the cosine schedule [39] with number of timestep $T = 25$. The model is trained up to 50 epochs by default on an NVIDIA 3090. We used the same training

settings as ProteinMPNN [6], where the batch size was set to approximately 6000 residues, and Adam optimizer [30] with noam learning rate scheduler [51] was used.

**Baselines**    We compare Bridge-IF with several state-of-the-art baselines, categorized into three groups: (1) autoregressive models, including StructGNN [25], GraphTrans [25], GCA [48], GVP [27], AlphaDesign [11], ESM-IF [22], and ProteinMPNN [6]; (2) the one-shot model, PiFold [12]; (3) iterative models, including LM-Design [64], KW-Design [14], and diffusion-based GraDe-IF [58].

**Evaluation**    We evaluate the generative quality using *perplexity* and *recovery rate*. Following previous studies [25, 22], we report perplexity and median recovery rate on three settings, namely short proteins (length ≤ 100), single-chain proteins (labeled with 1 chain in CATH), and all proteins.

## 5.2    Inverse folding

The performance of Bridge-IF, compared to competitive baselines, is summarized in Table 1. Bridge-IF demonstrates superior performance over previous methods. We highlight the following: (1) Iterative models comprehensively surpass the previously dominant autoregressive and one-shot methods. (2) Our Bridge-IF outperforms LM-Design and KW-Design with the same pre-trained PLMs, supporting our hypothesis that the iterative refinement process should be modeled in a probabilistic framework. (3) Compared with diffusion-based GraDe-IF, our Bridge-IF achieves better performance with fewer diffusion steps (25 vs. 500), demonstrating that our bridge-based formulation can better leverage the structural prior.

Following Zheng et al. [64], we also study the impact of the scale of PLMs on CATH v4.3. We use ESM-2 series, with parameters ranging from 8M to 3B. As depicted in Figure 3, the performance of Bridge-IF improves with model scaling, exhibiting a distinct scaling law in logarithmic scale. Using ESM-2 at the same scale, we observe that Bridge-IF consistently obtains greater enhancements relative to LM-Design. Besides, Bridge-IF does not exhibit any performance degradation, even when the smallest model (i.e, ESM-2 8M) is employed. Remarkably, the largest ESM2-3B-based variant of Bridge-IF attains a record-setting recovery rate of 61.27% on CATH v4.3.

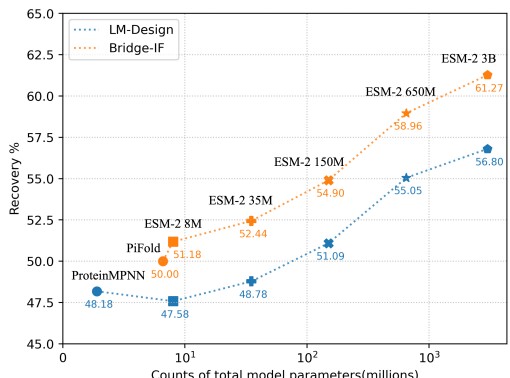

Figure 3: Performance comparison w.r.t. model scales of pLMs using ESM-2 series on CATH 4.3.

## 5.3    Foldability

While perplexity and recovery rate serve as effective proxy metrics, it is imperative to recognize that these measurements may not accurately reflect the foldability of the designed protein sequences in real-world scenarios [58, 13, 53]. Given that wet-lab assessment is extremely costly, we leverage the *in silico* structure prediction model ESMFold [34], to evaluate whether our designs can adhere to the structure condition. Here we assess the agreement of the native structures with the predicted structures using the TM-score [61], and follow the evaluation configurations as in Wang et al. [53]. Specifically, we use the small, high-quality test set of 82 samples curated by Wang et al. [53] and randomly generate 100 sequences for each structure.

Table 2: Numerical comparison on foldability and recovery rate. Benchmarked results are quoted from Wang et al. [53]. The **best** and suboptimal results are labeled with bold and underline.

| Model | TM-score | Recovery % |
|-------|----------|------------|
| Native sequences | 0.80 | 100.00 |
| Uniform | 0.05 | 5.00 |
| Natural frequencies | 0.07 | 5.84 |
| GraphTrans | 0.72 | 35.89 |
| GVP | 0.73 | 39.46 |
| ProteinMPNN | 0.80 | 41.44 |
| PiFold | 0.71 | 44.86 |
| LM-Design | 0.73 | 51.23 |
| Bridge-IF | **0.81** | **54.08** |

We report the TM-score and recovery metrics in Table 2. We observe that our Bridge-IF stands out as the leading model, exhibiting both high

**PDB ID: 1t07.A**          **PDB ID: 3ffv.A**          **PDB ID: 1uc2.B**

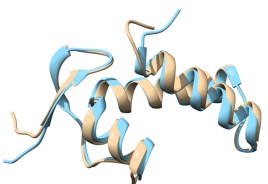          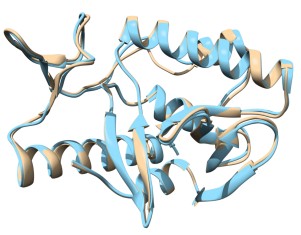          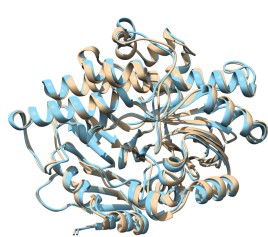

Recovery Rate: 0.69          Recovery Rate : 0.65          Recovery Rate : 0.76
TM-score: 0.91               TM-score : 0.95               TM-score : 0.99

Figure 4: Folding comparison of our designed sequences (in blue) and the native sequences (in nude).

Table 3: Ablation studies of key design choices on CATH v4.2. "w/ AdaLN-Bias" replaces the vanilla AdaLN with AdaLN-Bias. "w/ SCE" replaces the variational lower bound loss with simplified cross-entropy loss.

| Prior | Architecture | Objective | Perplexity ↓ | | | Recovery Rate % ↑ | | |
|---|---|---|---|---|---|---|---|---|
| w/ pre-training | w/ AdaLN-Bias | w/ SCE | Short | Single-chain | All | Short | Single-chain | All |
| | ✓ | ✓ | 6.51 | 6.30 | 4.23 | 43.17 | 44.29 | 56.53 |
| ✓ | | ✓ | 5.98 | 5.27 | 3.89 | 43.45 | 48.01 | 57.92 |
| ✓ | ✓ | | 6.52 | 6.40 | 4.28 | 43.43 | 44.01 | 56.43 |
| ✓ | ✓ | ✓ | **5.68** | **5.06** | **3.83** | **43.86** | **48.96** | **58.59** |

foldability and a high recovery rate. Notably, the predicted structures of our redesigned sequences align more closely with the given structures than do the native sequences, implying better structural validity of our redesigns. Another interesting finding is that PiFold and LM-Design achieve high recovery via a discriminative formulation but fall short on TM-score, indicating the limitation of structure-agnostic metrics. In contrast, probabilistic models Bridge-IF and ProteinMPNN,[2] perform exceptionally well on foldability. These results support our hypothesis that inverse protein folding should be modeled in a probabilistic framework considering the absence of a unique native sequence for a given backbone structure. Figure 4 showcases several instances where the folded structures of sequences designed by Bridge-IF are compared with reference crystal structures.

### 5.4 *De novo* protein design

Thus far, our experiments have been limited to accurate experimentally-determined structures. However, in real-world applications like *de novo* protein design, inverse folding models are commonly used to design sequences for novel structures generated by backbone generation models [56, 26]. Consequently, we next evaluate Bridge-IF for its potential in such a scenario. The experimental methodology is detailed as follows: we sample 10 backbones at every length $[100, 105, \ldots, 500]$ in intervals of 5 using Chroma [26]. For each de novo structure, we employ inverse folding models to design 8 sequences. Subsequently, these sequences are folded using ESMFold to identify the sequence with the highest TM-score (scTM). We compare Bridge-IF with ProteinMPNN [6], which is widely used in *de novo* protein design [59, 57]. Our results show that Bridge-IF surpasses ProteinMPNN in terms of scTM (0.73 vs. 0.69) and designability (0.85 vs. 0.80), using scTM $> 0.5$ as the criterion.

### 5.5 Ablation Studies

We conduct ablation experiments on CATH v4.2 to verify the impact of key design choices, and present the results in Table 3.

---

[2]ProteinMPNN, with its order-agnostic modeling, can be viewed as an autoregressive diffusion model [20].

### 5.5.1 Prior

We investigate two training strategies distinguished by their prior: 1) the structure encoder and the PLM are jointly trained; 2) the structure encoder is first pre-trained and remains frozen during the subsequent training of the PLM. We noted that the structure encoder is trained with an equivalent objective in both strategies. The latter consistently yields higher-quality protein sequences. Hence, it has been established as our default configuration.

### 5.5.2 Training objective

We find that the proposed simplified cross-entropy loss works better than the variational lower bound loss [24], demonstrating that the inferior performance of the vanilla Markov bridge model may stem from a harder optimization.

### 5.5.3 Network architecture

We observe that the performance of Bridge-IF further increases ($57.92\% \rightarrow 58.59\%$) when we replace the vanilla AdaLN with the proposed variant AdaLN-Bias. We highlight the use of AdaLN-Bias to enhance compatibility with pre-trained parameters when modulating a pre-trained Transformer model with additional conditions.

## 6 Conclusion

In this work, we introduce Bridge-IF, the first diffusion bridge model based on the Markov bridge process for inverse protein folding. Bridge-IF can gradually generate high-quality protein sequences from a deterministic prior. Bridge-IF achieves state-of-the-art performance in sequence recovery and foldability. **Future work** will focus on investigating more advanced structural encoders [38] and pre-training Bridge-IF using more protein structure data predicted by AlphaFold2 [28] to further enhance performance. We also intend to apply Bridge-IF to guide protein engineering aimed at designing novel functional proteins. **One potential limitation** of the proposed Bridge-IF is its lack of validation through wet-lab experiments in practical applications.

## Acknowledgments and Disclosure of Funding

This research was partially supported by National Natural Science Foundation of China under grants No.12326612, Zhejiang Key R&D Program of China under grant No. 2023C03053 and No. 2024SSYS0026, Alibaba Research Intern Program.

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

# A  Algorithms

The overall workflow of the training and sampling process are provided in Algorithm 1 and Algorithm 2.

---

**Algorithm 1** Training of the Bridge-IF

---

**Input:** coupled sample $(s, y) \sim p_{\mathcal{S}, \mathcal{Y}}$, structure encoder $\mathcal{E}$, neural network $\varphi_\theta$
$x = \mathcal{E}(s)$                  ▷ Deterministic mapping from structure to sequence
$t \sim \mathcal{U}(0, \ldots, T-1), \; z_t \sim \mathrm{Cat}\left(z_t; \overline{Q}_{t-1} x\right)$     ▷ Sample time step and intermediate state
$\hat{y} \leftarrow \varphi_\theta(z_t, t)$             ▷ Output of $\varphi_\theta$ is a vector of probabilities
$p(z_{t+1}|z_t, y) \leftarrow \mathrm{Cat}\left(z_{t+1}; Q_t(y) z_t\right)$     ▷ Reference transition distribution
$q_\theta(z_{t+1}|z_t) \leftarrow \mathrm{Cat}\left(z_{t+1}; Q_t(\hat{y}) z_t\right)$     ▷ Approximated transition distribution
Minimize $D_{\mathrm{KL}}\left(p(z_{t+1}|z_t, y) \| q_\theta(z_{t+1}|z_t)\right)$

---

**Algorithm 2** Sampling

---

**Input:** starting point $s \sim p_{\mathcal{S}}$, structure encoder $\mathcal{E}$, neural network $\varphi_\theta$
$z_0 \leftarrow \mathcal{E}(s)$
**for** $t$ in $0, \ldots, T-1$:
    $\hat{y} \leftarrow \varphi_\theta(z_t, t)$           ▷ Output of $\varphi_\theta$ is a vector of probabilities
    $q_\theta(z_{t+1}|z_t) \leftarrow \mathrm{Cat}\left(z_{t+1}; Q_t(\hat{y}) z_t\right)$     ▷ Approximated transition distribution
    $z_{t+1} \sim q_\theta(z_{t+1}|z_t)$
Return $z_T$

---

# B  Additional results

## B.1  Multi-chain protein complex design

Studying protein sequence design for multi-chain assemble structures is crucial for drug design. Next, we assess the capabilities of designing multi-chain complexes using the PDB dataset curated by Dauparas et al. [6], where sequences were clustered at 30% identity, resulting in 25,361 clusters. Following the standard data splitting, we divided those clusters randomly into three groups for training (23,358), validation (1,464), ensuring that neither the chains from the target chain nor the chains from the biounits of the target chain would be present in the other two groups.

As shown in Table 4, Bridge-IF also achieves similar improvements when extending to the PDB dataset, further validating its effectiveness and generalizability.

Table 4: Performance on multi-chain protein complex dataset (in median recovery). Results of the original ProteinMPNN and GVP-Transformer were obtained using publicly available checkpoints.

| Models | Rec. (↑) |
|---|---|
| ProteinMPNN [6] | 50.00 |
| ProteinMPNN + CMLM [ProtMPNN-CMLM] | 54.39 |
| LM-Design (ProtMPNN-CMLM + ESM-1b 650M) | 59.10 |
| LM-Design (pretrained ProtMPNN-CMLM: *freeze*) | 59.43 |
| LM-Design (pretrained ProtMPNN-CMLM: *fine-tune*) | 59.43 |
| LM-Design (ProtMPNN-CMLM + ESM-2 650M) | 59.81 |
| Bridge-IF (pretrained PiFold:*freeze* + ESM-2 650M) | 61.26 |

These results show that Bridge-IF can not only design single-chain proteins, which are mostly studied in previous works but also be used for designing multi-chain protein complexes.

## C  Derivations for the variational bound of reparameterized Markov bridge models

We derive the variational bound on negative log-likelihood $\log q_\theta(\boldsymbol{y}|\boldsymbol{x})$ as discussed in Section 4.2.

$$
\begin{aligned}
-\log q_\theta(\boldsymbol{y}|\boldsymbol{x}) &= -\log q_\theta(\boldsymbol{z}_T|\boldsymbol{z}_0) \\
&= -\log \int q_\theta(\boldsymbol{z}_{1:T}, v_{1:T}|\boldsymbol{z}_0) \, dv_{1:T} \, d\boldsymbol{z}_{1:T-1} \\
&= -\log \int \frac{p(\boldsymbol{z}_{1:T}, v_{1:T}|\boldsymbol{z}_0, \boldsymbol{z}_T)}{p(\boldsymbol{z}_{1:T}, v_{1:T}|\boldsymbol{z}_0, \boldsymbol{z}_T)} q_\theta(\boldsymbol{z}_{1:T}, v_{1:T}|\boldsymbol{z}_0) \, dv_{1:T} \, d\boldsymbol{z}_{1:T-1} \\
&\leq -\int p(\boldsymbol{z}_{1:T}, v_{1:T}|\boldsymbol{z}_0, \boldsymbol{z}_T) \log \frac{q_\theta(\boldsymbol{z}_{1:T}, v_{1:T}|\boldsymbol{z}_0)}{p(\boldsymbol{z}_{1:T}, v_{1:T}|\boldsymbol{z}_0, \boldsymbol{z}_T)} \, dv_{1:T} \, d\boldsymbol{z}_{1:T-1} \\
&= T \cdot \mathbb{E}_{t \sim \mathcal{U}(0,\dots,T-1)} \mathcal{L}_t(\theta)
\end{aligned}
$$

where

$$
\begin{aligned}
&\mathcal{L}_t(\theta) \\
&= \mathbb{E}_{p(\boldsymbol{z}_t|\boldsymbol{x},\boldsymbol{y})} \left[ \mathbb{E}_{p(\boldsymbol{v}_t)}[\mathrm{KL}(p(\boldsymbol{z}_{t+1}|v_t, \boldsymbol{z}_t, \boldsymbol{z}_T)||q_\theta(\boldsymbol{z}_{t+1}|v_t, \boldsymbol{z}_t))] + \mathrm{KL}\left(p(v_t)||q_\theta(v_t)\right) \right].
\end{aligned}
$$

We adopt the simplifying assumption that $q_\theta(v_t) = p(v_t)$, then $\mathcal{L}_t(\theta)$ can be written as

$$
\mathcal{L}_t(\theta) = \mathbb{E}_{p(\boldsymbol{z}_t|\boldsymbol{x},\boldsymbol{y})p(v_t)}[\mathrm{KL}(p(\boldsymbol{z}_{t+1}|v_t, \boldsymbol{z}_t, \boldsymbol{z}_T)||q_\theta(\boldsymbol{z}_{t+1}|v_t, \boldsymbol{z}_t))], \tag{11}
$$

in which the KL divergence has the form

$$
\mathrm{KL}[p(\boldsymbol{z}_{t+1}|v_t, \boldsymbol{z}_t, \boldsymbol{z}_T)||q_\theta(\boldsymbol{z}_{t+1}|v_t, \boldsymbol{z}_t)] = \begin{cases} (1 - \beta_t)\mathrm{KL}(\boldsymbol{y}||\varphi_\theta(\boldsymbol{z}_t, t)) & \text{if } v_t = 1 \\ \mathrm{KL}(\boldsymbol{z}_t||\boldsymbol{z}_t) = 0 & \text{if } v_t = 0 \end{cases} \tag{12}
$$

Given that $\mathrm{KL}(\boldsymbol{y}||\varphi_\theta(\boldsymbol{z}_t, t)) = -\boldsymbol{y}^T \log \varphi_\theta(\boldsymbol{z}_t, t)$, we have

$$
\begin{aligned}
&\mathbb{E}_{p(v_t)}[\mathrm{KL}\left[(p(\boldsymbol{z}_{t+1}|v_t, \boldsymbol{z}_t, \boldsymbol{z}_T)||q_\theta(\boldsymbol{z}_{t+1}|v_t, \boldsymbol{z}_t))\right] \\
&= -(1 - \beta_t)v_t \boldsymbol{y}^T \log \varphi_\theta(\boldsymbol{z}_t, t)
\end{aligned}
$$

## D  Broader impacts

Inverse protein folding models, operating within the broader realm of bioinformatics and computational biology, have significant impacts across various scientific and practical domains. These models, by enabling the design or prediction of protein sequences that fold into specific three-dimensional structures, foster advancements in numerous fields. The broader impacts encompass several areas, including drug discovery, enzyme design, and synthetic biology.

