# OpenReview forum: "Bridge-IF: Learning Inverse Protein Folding with Markov Bridges"
_NeurIPS.cc/2024/Conference — NeurIPS 2024 poster_

### Official Review · Reviewer_HBUp · 2024-07-07

**Soundness:** 3
**Presentation:** 3
**Contribution:** 2
**Rating:** 6
**Confidence:** 3

**Summary:**

This work introduces the bridge process generative model for inverse protein folding using Markov bridges to learn the probabilistic dependencies between protein backbone structures and sequences. This model aims to address the limitations of existing methods, such as error accumulation and the one-to-many mapping nature of inverse folding. Bridge-IF leverages an expressive structure encoder and integrates structural conditions into protein language models, achieving state-of-the-art accuracy on standard benchmarks.

**Strengths:**

[+] The use of Markov bridges for inverse protein folding is novel. The method and background is well-introduced.
[+] The Bridge-IF demonstrates superior performance in sequence recovery compared to existing baselines.
[+] The integration of structural conditions into protein language models significantly uses pre-trained information, and therefore improves generation performance while maintaining parameter efficiency.

**Weaknesses:**

[-] While the authors show the accuracy / PPL and TM-scores, I wonder whether it is possible to show that, experimentally, the proposed method can make discovery of novel protein sequences, or engineer existing proteins. The GFP can be a common benchmark.
[-] Instead of common benchmarks, I wonder whether the proposed method can show improvements on highly complex protein structures.
[-] While introducing the background of bridges, I think the authors can do a more details discussion and introductions, e.g., including some missing references published in last two years Neurips / ICML conferences.

**Questions:**

Please refer to the weaknesses part.

**Limitations:**

I do not think this work has potential negative societal impact.

---

> ### Author Rebuttal · Authors · 2024-08-07
>
> We deeply appreciate the reviewer for the insightful and constructive comments!
>
> > While the authors show the accuracy / PPL and TM-scores, I wonder whether it is possible to show that, experimentally, the proposed method can make discovery of novel protein sequences, or engineer existing proteins. The GFP can be a common benchmark.
>
> Thanks for your constructive comment. As wet-experimental assessments are exceptionally costly and time-consuming, we are currently unable to verify Bridge-IF’s ability to discover novel proteins. Nevertheless, we can leverage the zero-shot mutation effect predictive power of Bridge-IF to engineer existing proteins. Specifically, we score the mutants by conditioning the sequence likelihoods, measured as the evidence lower bound in Equation 7, on a known structure. We then compute the Spearman’s rank correlation between the model scores and the corresponding experimental measurements. On the common GFP benchmark [1], our Bridge-IF outperforms state-of-the-art inverse folding models, including ProteinMPNN and ESM-IF. The high correlation suggests that our model is likely to identify variants with enhanced fitness in real-world applications [2].
>
> || Spearman’s rank correlation |
> |-|-|
> | ProteinMPNN | 0.602 |
> | ESM-IF | 0.713 |
> | Bridge-IF | 0.722 |
>
> [1] Sarkisyan, Karen S., et al. "Local fitness landscape of the green fluorescent protein." Nature 533.7603 (2016): 397-401.
>
> [2] Hie, Brian L., et al. "Efficient evolution of human antibodies from general protein language models." Nature Biotechnology 42.2 (2024): 275-283.
>
> > Instead of common benchmarks, I wonder whether the proposed method can show improvements on highly complex protein structures.
>
> Our Bridge-IF can achieve promising performance on the design of multi-chain protein complexes and de novo proteins. As shown in Appendix B.1, Bridge-IF outperforms ProteinMPNN and LM-Design on designing multi-chain proteins with highly complex structures. In addition, we also conduct experiments on de novo protein design, where inverse folding models are used to design sequence for structures generated by the structure generation model. Specifically, we sample 10 backbones at every length [100, 105, . . . , 495, 500] in intervals of 5 (810 total samples) using Chroma [3], and generate 8 sequences for each backbone using different inverse folding methods. Then these sequences were folded using ESMFold to select the sequence with the highest TM-score (sc-TM). We found that Bridge-IF outperforms ProteinMPNN in terms of average sc-TM and Designability (sc-TM > 0.5). We believe that these promising results demonstrate that Bridge-IF can generalize better to more complex structures and be applied to the design of multi-chain protein complexes and de novo proteins.
>
> || scTM | designability (scTM > 0.5)
> |-|-|-|
> | ProteinMPNN  | 0.69 | 0.80 |
> | Bridge-IF | 0.73 | 0.85 |
>
> [3] Ingraham, John B., et al. "Illuminating protein space with a programmable generative model." Nature 623.7989 (2023): 1070-1078.
>
> > While introducing the background of bridges, I think the authors can do a more details discussion and introductions, e.g., including some missing references published in last two years Neurips / ICML conferences.
>
> Thanks for pointing this out. We will revise the related work to include more important references, to name a few [4,5,6], and offer a more detailed introduction.
>
> [4] Liu, Xingchao, et al. "Let us build bridges: Understanding and extending diffusion generative models." arXiv preprint arXiv:2208.14699 (2022).
>
> [5] Zhou, Linqi, et al. "Denoising Diffusion Bridge Models." The Twelfth ICLR.
>
> [6] Shi, Yuyang, et al. "Diffusion Schrödinger bridge matching." NeurIPS 36 (2023).
>
> **We hope our response can alleviate your concerns. Please let us know if you have any additional questions.**

---

> > ### Comment · Reviewer_HBUp · 2024-08-13
> > **Reponse**
> >
> > I thank the authors for their response. The related work discussion will be helpful and the additional experiments are useful for understanding the model performance. I will keep my weak accept score.

---

> > > ### Author Response · Authors · 2024-08-14
> > >
> > > We sincerely appreciate your valuable comments which greatly help us to improve our work and will incorporate the discussions and results into the final version.

---

### Official Review · Reviewer_r7sf · 2024-07-09

**Soundness:** 2
**Presentation:** 3
**Contribution:** 3
**Rating:** 6
**Confidence:** 5

**Summary:**

The authors propose a Markov Bridge-based model for the protein inverse folding problem, incorporating a language model. This model outperforms all baseline methods on the CATH dataset. However, achieving state-of-the-art results on accurate crystal test datasets alone is not sufficient in practice. In real-world applications, protein inverse folding is often applied to structures with noise (e.g., structures generated by diffusion or predicted by software like AlphaFold).

I expect the authors to add relevant experiments to illustrate the generalizability of the method.

**Strengths:**

1. The authors used the Markov bridge method for sequence design and incorporated a language model that demonstrated superior performance on the CATH dataset.

2. I was pleased to see that the authors explored the performance of complex sequence design, and Bridge-IF also achieved superior performance on the complex dataset.

**Weaknesses:**

**Major**

Protein inverse-folding software is crucial for de novo protein design. Experiments in Bridge-IF focus only on accurate experimentally determined structures, which does not prove that Bridge-IF generalizes better than other methods. In real-world applications, designs are often made for structures generated by de novo software or predicted by structure prediction software. Complementing inverse folding experiments on de novo proteins is essential. The experiment details are as follows:

Proteins of lengths 100-500 were designed using de novo methods (e.g., RFdiffusion or Chroma). For each de novo protein, 8 sequences were designed using each method, and then these sequences were folded using ESMFold to select the sequence with the highest TM-score.

Considering the rebuttal time constraints, showing a de novo design approach and comparing two of the strongest current baselines is sufficient to illustrate the superiority of Bridge-IF. These baselines are ChromaDesign (t=0.5 version) (https://github.com/generatebio/chroma/tree/main) and ProteinMPNN (input Ca atoms only, v_48_020 version) (https://github.com/dauparas/ProteinMPNN/tree/main/ca_model_weights).

I would be happy to increase my score if Bridge-IF achieves significant improvements on the above experiments.

**Minor**

1. Lack of novelty. Using diffusion models and protein language models to design protein sequences is not new. This work seems to combine previous works (LM-Design, GraDe-IF) and make incremental changes.

2. The improvement (1-2%) in recovery rate is limited.

3. Lacks many baselines. The core motivation of Bridge-IF is to enhance the quality of samples. Although many inverse-folding methods are orthogonal to Bridge-IF, a detailed comparison is needed. Like SPDesign (http://zhanglab-bioinf.com/SPDesign/), ChromaDesign (t=0 version on CATH dataset) (https://github.com/generatebio/chroma/tree/main) , and InstructPLM (https://github.com/Eikor/InstructPLM?tab=readme-ov-file).

4. The authors state that the discriminant model cannot handle the one-to-many case between protein sequence and structure.
However, ProteinMPNN and ESM-IF provide sampling temperatures for generating diverse sequences, and ESM3 (also based on a discriminative model) can generate diverse structures and sequences. The article does not have a corresponding benchmark for why the diffusion-based model is better at handling one-to-many situations.

6. ProteinMPNN provides many versions of models. If the authors want to compare recovery rates on experimentally determined structures, they should select the backbone atom input, add a model with little noise (0.02), and set the sampling temperature to 1e-5.

**Questions:**

1. ESM-IF does not release a version that is not trained with AlphaFold DataBase. How did the authors evaluate this?

2. According to my statistics, the CATH dataset has many sequences in the UniRef database, while ESM-1b and ESM-2 were trained using UniRef datasets, respectively. How do the authors deal with potential information leakage?

3. Discriminative methods for sequence design based on the Potts model (like ChromaDesign [1] or CarbonDesign [2]) can also alleviate the problem of "Error accumulation" by sampling the amino acid pairs using a gradient-based or a multistep greedy method. Is the conclusion in line 32 too strong? I would like the author to add a relevant discussion.

4. The authors mention "accelerate the inference process" in the motivation, but there is no comparative reasoning time. They used a pre-trained language model, a more complex network. Can the inference process be accelerated (especially compared to auto-regressive-based methods)?

5. The TMscore of Bridge-IF's designed sequence exceeds that of the natural sequence. Does this indicate that the structure prediction software prefers Bridge-IF designed sequences? I would like to see a discussion from the authors about this point.

[1] Illuminating protein space with a programmable generative model. Nature, 2023.

[2] Accurate and robust protein sequence design with CarbonDesign. Nature Machine Intelligence, 2024.

**Limitations:**

The authors address limitations and potential negative societal impact.

---

> ### Author Rebuttal · Authors · 2024-08-07
>
> We deeply appreciate the reviewer for the insightful and constructive comments!
>
> > Experiment on de novo proteins.
>
> We conducted experiments based on your suggested setups. We sample 10 backbones at every length [100, 500] in intervals of 5 using Chroma. Inspired by the finding of ProteinMPNN that training with noise improves performance for real applications, we trained a Bridge-IF with 0.2-Å noise on the dataset curated by ProteinMPNN for a fair comparison.
>
> Note that ChromaDesign is trained on the same training structure data as Chroma, it is not surprising that ChromaDesign achieves excellent performance in predicting sequences for structures generated by Chroma. In contrast, Bridge-IF and ProteinMPNN are trained on the same data and tested in the same manner, leading to a fair comparison. We found that Bridge-IF outperforms ProteinMPNN (Ca_only, v_48_020) in terms of the highest TM-score (scTM) and designability (scTM > 0.5). Because ProteinMPNN is widely used in de novo protein design and evaluating backbone generation models, we believe that Bridge-IF can generalize better to de novo structures and be applied to de novo design. Detailed distribution of the TM-score w.r.t. sequence length is presented in the PDF for global response.
>
> ||scTM| designability (scTM > 0.5)
> |-|-|-|
> |ChromaDesign|0.86 | 0.98 |
> |ProteinMPNN| 0.69 | 0.80 |
> |Bridge-IF| 0.73 | 0.85 |
>
> > This work seems to combine previous works and make incremental changes.
>
> We emphasize that Bridge-IF is not simply a combination of diffusion models and pLMs.  **We have sufficient novelty and contributions in both algorithms and network architectures:**
>
> - We present the first bridge-based model for inverse folding and offer a reparametrization of the Markov bridge model, which enhances the training of the vanilla Markov bridge model (Section 4,2). Additionally, we have implemented a pLM-augmented GraDe-IF by substituting the denoising network with ESM-1b 650M. Despite this modification, our Bridge-IF continues to outperform GraDe-IF (Recovery: 58.49 v.s. 55.37), underscoring the advantages of Markov bridges over diffusion models.
> - Our backbone is different from LM-Design with additional timestep conditions. To integrate timestep, our proposed adaLN-Bias is also different from adaLN used in DiT, designed for better integration with pre-trained parameters of LN (Section 4.3).
>
> The effectiveness of the above design choices is verified with ablations (Section 5.4).
>
> > The improvement in recovery rate is limited.
>
> Recovery rate may not be the only criterion for evaluating the quality of the designed sequences. Bridge-IF is the first model that can accommodate both recovery and foldability. In Table 2, LM-Design can achieve performance closest to Bridge-IF on recovery, but the sequence foldability is significantly lower than that of Bridge-IF.
>
> > Lacks many baselines.
>
> For SPDesign, the data partitioning differs from all baselines, the source code is not available, and the server is out of service currently, preventing a fair comparison. For Chroma (t=0), the inconsistent training data and unavailable training code also make comparison hard. We evaluated on the intersection of the Chroma and CATH 4.2 test set, and found Bridge-IF outperforms Chroma significantly (57.52% v.s. 51.28%). Compared with InstructPLM, Bridge-IF achieves a higher recovery rate (58.59 v.s. 57.51), despite using a smaller pLM (ESM-2 650M v.s. ProGen2 6.4B). We will add this discussion for a detailed comparison.
>
> > Benchmark for diversity.
>
> We are sorry for not conducting a comparison of diversity due to the lack of benchmarks. For de novo protein design, we compute the diversity of 8 sequences generated for the same backbone according to [1], and find that Bridge-IF generates more diverse samples.
>
> ||diversity|
> |-|-|
> |ChromaDesign|0.34|
> |ProteinMPNN|0.44|
> |Bridge-IF|0.56|
>
> [1] Biological sequence design with gflownets.
>
> > ESM-IF does not release a version that is not trained with AFDB.
>
> The results of ESM-IF on CATH 4.3 are quoted from Table 1 in the original paper.
>
> > Potential information leakage.
>
> We remark that recent works have leveraged pLMs for protein inverse folding. Following these works, we used the same PLM and evaluated on the standard benchmark without additional data processing. Besides, the promising performance on de novo protein design verified Bridge-IF's ability to design novel sequences that were not present in its training data.
>
> > Potts model can alleviate error accumulation.
>
> We use discriminative formulation to emphasize that autoregressive and one-shot models formulate the problem as the prediction of the most probable sequence for a given protein structure. Hence, we do not want to include Potts model, whose sampling process is similar to diffusion models. We will add a relevant discussion about potts model.
>
> > Can the inference be accelerated, compared to autoregressive methods (ARM)?
>
> The complexity of ARM is O(L) to the protein length L, while that of bridge/diffusion-based models is O(T) to the total timestep T. This fact demonstrates that our Bridge-IF can work with fewer number of function evaluations (NFE) than ARM, especially for larger proteins. Compared with GraDe-IF (500 NFE), Bridge-IF achieves better performance with fewer diffusion steps (25 NFE). We profiled the inference speed on a NVIDIA RTX 3090, averaging over the sampling time for sequences generated by Chroma.
>
> | Length|100-200|200-300|300-400|400-500|
> |-|-|-|-|-|
> | ChromaDesign | 4.67s| 6.74s| 8.24s| 10.29s|
> | ProteinMPNN | 1.52s | 4.52s|7.45s|13.03s|
> | Bridge-IF|1.04s| 1.56s|1.98s| 2.30s|
>
> > The TM-score of Bridge-IF's designed sequence exceeds natural sequences.
>
> This phenomenon is also observed in the original papers of ProteinMPNN. We reasoned that Bridge-IF might design sequences for native backbones that better encode their structures, because evolution does not always prioritize structural stability. We will add this discussion in Section 5.3.

---

> > ### Comment · Reviewer_r7sf · 2024-08-08
> >
> > I'm glad to see that the author's response has addressed most of my questions and I changed the score.
> >
> > Bridge-IF surpasses ProteinMPNN in de novo protein sequence design, suggesting Bridge-IF has more potential for future exploration.
> >
> > However, I still have some questions, and if the authors can continue to address them, I would be willing to continue to raise my score.
> > 1. "Note that ChromaDesign is trained on the same training structure data as Chroma, it is not surprising that ChromaDesign achieves excellent performance in predicting sequences for structures generated by Chroma." is not accurate, ChromaDesign is trained separately, and I think the main reason for this is the noise added for the training data, not the adaptive nature of Chroma and ChromaDesign. I have tried ChromaDesign which is very robust for data generated by different models.
> > The authors could try using one of RFDiffusion (https://github.com/RosettaCommons/RFdiffusion/tree/main), FrameDiff (https://github.com/Immortals-33/Framediff/ tree/master), MultiFlow (https://github.com/jasonkyuyim/multiflow) or Proteus (https://github.com/Wangchentong/Proteus) to evaluate ChromaDesign and Bridge-IF.
> >
> > Notion: ChromaDesign makes a lot of improvements in de novo protein sequences, I think a fair comparison and illustration of potential enhancements is meaningful to the community, beyond ProteinMPNN is good enough, Bridge-IF doesn't have to achieve SOTA on all datasets.
> >
> > 2.The authors state that the Potts-model based approach has no error accumulation problems. Then the Potts-model approach is still a potential baseline as CarbonDesign (default) (https://github.com/zhanghaicang/carbonmatrix_public), and I would like the authors to add a relevant comparison in the CATH dataset. I would like to see more discussion from the author about the Potts model.
> >
> > 3.The authors should add related work on protein sequence design such as SPDesign, ChromaDesign, CarbonDesign, InstructPLM, etc.
> >
> > 4.I observed that the performance of Bridge-IF on long sequences is not as good as ChromaDesign, how to explain this phenomenon? Does it stem from a large crop size in ChromaDesign?

---

> ### Author Response · Authors · 2024-08-09
> **Response about de novo protein design**
>
> We deeply appreciate the reviewer for the detailed feedback which greatly helps us to improve our work.
>
> > "Note that ChromaDesign is trained on the same training structure data as Chroma, it is not surprising that ChromaDesign achieves excellent performance in predicting sequences for structures generated by Chroma." is not accurate, ChromaDesign is trained separately, and I think the main reason for this is the noise added for the training data, not the adaptive nature of Chroma and ChromaDesign. I have tried ChromaDesign which is very robust for data generated by different models. The authors could try using one of RFDiffusion, FrameDiff, MultiFlow or Proteus to evaluate ChromaDesign and Bridge-IF.
> >
> > Notion: ChromaDesign makes a lot of improvements in de novo protein sequences, I think a fair comparison and illustration of potential enhancements is meaningful to the community, beyond ProteinMPNN is good enough, Bridge-IF doesn't have to achieve SOTA on all datasets.
>
> We appreciate the opportunity to further dive into the application of the inverse folding model for de novo protein design. Below, we present our findings:
>
> - A trade-off exists between diversity and sample quality. We have tested ChromaDesign at varying temperature values (temperature_S). Note that the performance reported in our previous rebuttal is obtained with a default temperature value of 0.01. We observed that as temperature_S increases, diversity rises, while both scTM and designability decrease. Remarkably, when the diversity of the generated samples is comparable, the sample quality of Bridge-IF (scTM: 0.73, designability: 0.85, diversity: 0.56) surpasses that of ChromaDesign (temperature_S = 0.25).
>
> | temperature_S | scTM | designability (scTM > 0.5) | diversity |
> | -| - | - | - |
> | 0.01     | 0.86     | 0.98 | 0.34 |
> | 0.1      | 0.85     | 0.98 | 0.38 |
> | **0.25** | **0.70** | **0.83** | **0.53** |
> | 0.5      | 0.64     | 0.74 | 0.72 |
> | 1        | 0.42     | 0.18 | 0.84 |
>
> - Inspired by these results, we performed further comparisons between Bridge-IF with a lower sampling temperature and ChromaDesign using backbone structures generated by FrameDiff. As anticipated, we found that Bridge-IF at a lower temperature significantly improved its performance, yielding results comparable to ChromaDesign (temperature_S = 0.01) regarding scTM and designability (scTM > 0.5). Following suggestions from RFdiffusion and FrameDiff, we employed designability (scRMSD < 2Å) as a stricter metric, revealing that Bridge-IF achieves superior performance.
>
> |  | scTM | designability (scTM > 0.5) | designability (scRMSD < 2Å)
> | - | - | - | - |
> | ChromaDesign  | 0.86  | 0.99  | 0.41 |
> | Bridge-IF     | 0.86     | 0.99  | **0.44** |
>
> - We agree that the addition of noise to the training data is crucial for robust de novo protein design. For instance, ProteinMPNN systematically examines the impact of noise levels on sequence recovery and robust design. ChromaDesign utilizes diffusion augmentation, where the sequence prediction is based on a noisy structure and a specific time t, Given that ChromaDesign is conditioned on the noise level, it may provide enhanced flexibility in selecting the optimal noise level for particular design tasks. Due to time constraints during the rebuttal process, we could not conduct exhaustive attempts but achieved promising preliminary results. We will undertake further investigations in de novo protein design in future work.
>
> > I observed that the performance of Bridge-IF on long sequences is not as good as ChromaDesign, how to explain this phenomenon? Does it stem from a large crop size in ChromaDesign?
>
> Thanks for the careful review. While we also suspect that the large crop size in ChromaDesign may be a potential reason, we also conducted additional analyses. As noted in our previous response, a trade-off exists between diversity and sample quality. We argue that the suboptimal performance of our Bridge-IF on long sequences primarily results from insufficient control of the sampling temperature. This implementation leads to the generation of more diverse samples, which compromises performance on the more challenging long sequences. We found that ChromaDesign (temperature_S = 0.25), which produces samples demonstrating diversity comparable to that generated by Bridge-IF, also experiences a significant decline in performance (the highest TM-score) on length sequences, thereby validating our explanation.
>
> |  Length | 200-300 | 300-400 | 400-500 |
> | - | - | - | - |
> | ChromaDesign (temperature_S = 0.01) | 0.88 | 0.87 | 0.81 |
> | ChromaDesign (temperature_S = 0.25) | 0.69 | 0.62 | 0.46 |
> | Bridge-IF | 0.76 | 0.68 | 0.53 |

---

> ### Author Response · Authors · 2024-08-09
> **Response about related work**
>
> > The authors state that the Potts-model based approach has no error accumulation problems. Then the Potts-model approach is still a potential baseline as CarbonDesign (default), and I would like the authors to add a relevant comparison in the CATH dataset.
>
> Thanks for the constructive comment. Unfortunately, the training script for CarbonDesign is unavailable, which complicates a fair comparison in the CATH dataset. Additionally, according to the description in the original paper, CarbonDesign was trained on PDB data prior to 2020. Given that the CATH4.2 dataset is partitioned based on PDB data from before 2019 [1], there is a potential risk of data leakage when directly validating CarbonDesign on the CATH4.2 test set. Therefore, we opted to test our Bridge-IF on the CASP15 test set for comparison with CarbonDesign. Our results indicate that Bridge-IF outperforms state-of-the-art baselines, including CarbonDesign.
>
> |  | recovery rate (the results of baselines are quoted from CarbonDesign)|
> | -------- | -------- |
> | ProteinMPNN_002 | 0.48 |
> | ESM-IF | 0.50 |
> | CarbonDesign | 0.54 |
> | Bridge-IF | **0.566** |
>
> [1] Generative models for graph-based protein design, 2019.
>
> > I would like to see more discussion from the author about the Potts model.
>
> The Potts model, a class of energy-based models, is a statistical mechanics framework utilized to describe interactions among spins on a lattice. Potts models have been applied to capture the sequence coevolution information of proteins [2,3]. Typically, the parameters of the Potts model are inferred using plmDCA [4] or bmDCA [5], and Markov Chain Monte Carlo (MCMC) sampling [6] can be employed to generate novel sequences. In the context of inverse folding, ChromaDesign factorizes the conditional distribution of sequences as a conditional Potts model with the likelihood given by
> $$
> p_\theta(s|x) = \frac{1}{Z(x,\Theta)} \exp \left( -\sum_i h_i(s_i;x) - \sum_{i<j} J_{ij}(s_i,s_j|x) \right),
> $$
> where the conditional fields $h_i(s_i;x)$ and conditional couplings $J_{ij}(s_i,s_j|x)$ are parameterized by the node and edge embeddings of the graph neural network, respectively. Similarly, CarbonDesign proposes Inverseformer to derive representations from the input backbone structures while conditioning fields and couplings on the learned single and paired representations, respectively.
>
> We contend that the strength of the Potts model lies in its explicit formulation of probability distributions that align with domain knowledge. However, its limitations include a restriction to modeling only second-order effects, a requirement for significantly more iterations of Monte Carlo sampling, and potential issues with slow mode mixing.
>
> [2] Hopf, Thomas A., et al. "Mutation effects predicted from sequence co-variation." Nature biotechnology.
>
> [3] Russ, William P., et al. "An evolution-based model for designing chorismate mutase enzymes." Science。
>
> [4] Ekeberg, Magnus, Tuomo Hartonen, and Erik Aurell. "Fast pseudolikelihood maximization for direct-coupling analysis of protein structure from many homologous amino-acid sequences." Journal of Computational Physics.
>
> [5] Figliuzzi, Matteo, Pierre Barrat-Charlaix, and Martin Weigt. "How pairwise coevolutionary models capture the collective residue variability in proteins?." Molecular biology and evolution 35.4 (2018): 1018-1027.
>
> [6] Grathwohl, Will, et al. "Oops i took a gradient: Scalable sampling for discrete distributions." International Conference on Machine Learning. PMLR, 2021.
>
> > The authors should add related work on protein sequence design such as SPDesign, ChromaDesign, CarbonDesign, InstructPLM, etc.
>
> We apologize for not providing a more thorough discussion of these related works in our initial rebuttal due to word limit constraints. Note that we have discussed Potts-based ChromaDesign and CarbonDesign in the previous response.
>
> - SPDesign emphasizes protein-specific physicochemical features. Specifically, it employs ultrafast shape recognition vectors to identify similar protein structures within an in-house database and extracts sequence profiles via structural alignment to enhance sequence prediction. In contrast, our Bridge-IF prioritizes advancements in probabilistic modeling.
>
> - InstructPLM aligns autoregressive protein language models, such as ProGen2, with pre-trained structural features to tackle the inverse folding problem. Conversely, our Bridge-IF utilizes a non-autoregressive iterative refinement approach. We contend that the evolutionary acquisition of proteins implies that each amino acid largely depends on its spatially nearest neighbors; thus, the left-to-right generation method may be sub-optimal for protein generation.
>
> We will incorporate these discussions into our final version.
>
> **We sincerely thank you once again for your help in improving the quality of our paper. We hope our response can alleviate your concerns. Please let us know if you have any additional questions.**

---

> > ### Comment · Reviewer_r7sf · 2024-08-10
> >
> > Thanks for your reply, the author has addressed most of my questions and I think the Bridge-IF experiment and results are complete.
> > **Therefore, I recommend acceptance of this article.** I have raised my score.
> >
> > I recommend that the authors to (1) add relevant work (2) add relevant discussion (3) add the experiments in de novo to the paper, which will help to improve the impact and completeness of Bridge-IF.
> >
> > Although the authors' responses are complete enough, for Question1, I still want to know under the standard experimental setup (100-500 lengths, using either 8 times or 1 sequence design method for each protein), for RFdiffusion or FrameDiff-designed proteins, the standard setups of ChromaDesign (t=0.01) and Bridge-IF (defult setting) performance comparison. Because in my use and review, ChromaDesign is very robust software in de novo protein sequence design.

---

> > > ### Author Response · Authors · 2024-08-11
> > >
> > > Thank you very much for your timely reply and recognition of our efforts!
> > >
> > > > Although the authors' responses are complete enough, for Question1, I still want to know under the standard experimental setup (100-500 lengths, using either 8 times or 1 sequence design method for each protein), for RFdiffusion or FrameDiff-designed proteins, the standard setups of ChromaDesign (t=0.01) and Bridge-IF (default setting) performance comparison. Because in my use and review, ChromaDesign is very robust software in de novo protein sequence design.
> > >
> > > The results are presented in the table below. We also found that ChromaDesign is a robust software capable of generalizing to the backbone structures generated by FrameDiff.
> > >
> > > |  | scTM | designability (scTM > 0.5) | designability (scRMSD < 2Å)
> > > | - | - | - | - |
> > > | ChromaDesign  (t=0.01)          | 0.86     | 0.99  | 0.41 |
> > > | Bridge-IF (low temperature)    | 0.86     | 0.99  | **0.44** |
> > > | Bridge-IF (default)                   | 0.81     | 0.90  | 0.36 |
> > >
> > > **We appreciate your valuable comments and will incorporate the discussions and results into the final version to improve the impact and completeness of Bridge-IF.**

---

> > > > ### Comment · Reviewer_r7sf · 2024-08-14
> > > >
> > > > Thanks to the author's response, I have no additional questions about Bridge-IF
> > > > I think the final version of the paper after the rebuttal process of Bridge-IF could have the following changes:
> > > >
> > > > (1) In the conclusion section, the authors only state further wet-experiment validation, and do not say what else can be explored in the machine learning direction, which is something that would have far-reaching implications for the machine learning community. For example, the authors can add some conclusions from the following aspects:
> > > >
> > > > i) The exploration of input features, both PiFold and SPDesign discuss and illustrate the input features further,
> > > >
> > > > ii) Pre-training features, I am concerned that KW-Design and SPDesign use protein structure pre-training models, whereas Bridge-IF uses pre-training of sequences only,
> > > >
> > > > iii) The tendency of sequence design to be a function of features, e.g. CarbonDesign performed experiments on DMS and ClinVar to illustrate the link between sequence design and functional proteins.
> > > >
> > > > iv) Interpretability: what exactly is the framework improvement of the diffusion model over autoregressive, and why can auto-regressive text-based generation achieve very good results while sequence-generated diffusion would be better?
> > > >
> > > > v) Scaling, ESM-IF uses the AlphaFold Database as a training set, while ESM3 uses a wider range of data sources, how to use more data and scaling is also a concern of the field
> > > >
> > > > (2)Presentation of experimental details: In CarbonDesign's comparison test, I found that not all CASP15 data used by CarbonDesign is publicly available, and there are new CASP15 data published after CarbonDesign's work, is the author's test set different from CarbonDesign? CarbonDesign demonstrated performance for orphan, and long sequences, and future work could explore these.
> > > >
> > > > (3)Clear conclusions from the Rebuttal process.
> > > >
> > > > (4) Considerable work has been done on protein sequence design, and the authors could consider summarising the Beidge-IF position more clearly in tabular form
> > > >
> > > > I would like to thank the authors for their responses again during the rebuttal process and their contributions to the field of protein design and machine learning applications, and I would recommend acceptance of this article.
> > > >
> > > > Best wishes!

---

> > > > > ### Author Response · Authors · 2024-08-14
> > > > >
> > > > > We sincerely appreciate your valuable comments which greatly help us to improve our work and will incorporate these discussions and results into the final version.

---

### Official Review · Reviewer_pPDQ · 2024-07-12

**Soundness:** 2
**Presentation:** 3
**Contribution:** 2
**Rating:** 3
**Confidence:** 4

**Summary:**

The paper proposes a new method called Bridge-IF for the problem of inverse protein folding. The main objective is to develop a diffusion bridge generative model called Bridge-IF that can generate high-quality protein sequences from a structure-aware prior.  The authors claim that they outperform state-of-the-art baselines on standard benchmarks like CATH in terms of sequence recovery rate and perplexity.

**Strengths:**

There are some strengths:
- The paper is clearly written.
- The consistent performance improvement across different model sizes.
- Figures are beautiful.

**Weaknesses:**

Weakness include:
- The authors have toned down the baseline results. For example, KWDesign reported recovery rates above 60% for CATH4.2 and CATH4.3 in the original paper. However, this paper drops the results to 58%. Why is this happening?
- Compared to KWDesign, the results are not state-of-the-art. And, more importantly, this may indicate that the model could not obtain additional benefit from the diffusion model.

**Questions:**

My questions are:
- When you use CATH4.2 and CATH4.3 for evaluating PLM-based methods, there is a risk of data leakage. Because the CATH dataset may have been seen during the training of ESM models. Have you considered this? And how do you address this issue?
- Why not report the original results of KWDesign? How does your setup differ from theirs?

**Limitations:**

The authors said: One potential limitation of the proposed Bridge-IF is its lack of validation through wet-lab experiments in practical applications

---

> ### Author Rebuttal · Authors · 2024-08-07
>
> We deeply appreciate your critical comments on our paper. However, we believe some of the concerns are caused by potential misunderstandings, and we hope that our response can address your concerns.
>
> > **W1**: The authors have toned down the baseline results. For example, KWDesign reported recovery rates above 60% for CATH4.2 and CATH4.3 in the original paper. However, this paper drops the results to 58%. Why is this happening?
> >
> > **W2**: Compared to KWDesign, the results are not state-of-the-art. And, more importantly, this may indicate that the model could not obtain additional benefit from the diffusion model.
> >
> > **Q2**: Why not report the original results of KWDesign? How does your setup differ from theirs?
>
> We apologize for any misunderstandings caused by our unclear descriptions. Indeed, we **did not tone down** the baseline results. We aim to conduct a fair comparison using the same protein language model (pLM) backbone (i.e., ESM-2 650M) to verify the effectiveness of the proposed Markov-bridge-based method. The results of KW-Design are **quoted from the original paper** (see Table 5, https://openreview.net/pdf?id=mpqMVWgqjn). Specifically, the best performance of KW-Design (recovery rate 60.77% on CATH 4.2, 60.38% on CATH 4.3) is achieved by combining the ESM-2 650M and ESM-IF, which is pre-trained on large-scale structure data, leading an unfair comparison. Besides, in addition to refining sequences with pLM, KW-Design also introduces several complex mechanisms, such as virtual multiple sequence alignment, a confidence-aware gated attention mechanism, and memory retrieval, to enhance the model's performance. In contrast, our Bridge-IF is more simple and effective. Remarkably, despite KW-Design using more complex techniques and knowledge pre-trained on structural data, Bridge-IF with ESM-2 3B still outperforms ensemble KW-Design (recovery rate: 61.27% v.s. 60.38%) on CATH 4.3, as shown in our Figure 3.
>
> > When you use CATH4.2 and CATH4.3 for evaluating PLM-based methods, there is a risk of data leakage. Because the CATH dataset may have been seen during the training of ESM models. Have you considered this? And how do you address this issue?
>
> First of all, we remark that several recent works, such as LM-Design and KW-Design, have leveraged pLMs for protein inverse folding. Following these works, we used the same PLM and evaluated the proposed methods on the standard benchmark without any additional data processing. Therefore, the comparison is fair.
>
> Here are our thoughts about data leakage:
> - As pLMs such as ESMs are trained solely on sequence data, they cannot directly be applied for the inverse folding problem on their own, regardless of whether or not the proteins are part of a pLM's training data. As shown in the preliminary study of LM-Design, the authors found that directly using pLMs to revise the predicted sequences can only yield a marginal enhancement. Hence, the focus of the recent research is on how to better utilize pLMs to facilitate the inverse folding task.
>
> - To further eliminate the potential concern of model memorization and ensure that our model is capable of generalizing, we conducted experiments on a set of de novo protein structures. Specifically, we sample 10 backbones at every length [100, 105, . . . , 495, 500] in intervals of 5 (810 total samples) using Chroma [1], and generate 8 sequences for each backbone using different inverse folding methods. Then these sequences were folded using ESMFold to select the sequence with the highest TM-score (sc-TM). We found that Bridge-IF outperforms ProteinMPNN in terms of average sc-TM and Designability (sc-TM > 0.5), verifying Bridge-IF's ability to design novel sequences for de novo protein structures that were not present in its training data.
>
> || scTM | designability (scTM > 0.5)
> |-|-|-|
> | ProteinMPNN  | 0.69 | 0.80 |
> | Bridge-IF | 0.73 | 0.85 |
>
> [1] Ingraham, John B., et al. "Illuminating protein space with a programmable generative model." Nature 623.7989 (2023): 1070-1078.
>
> **We hope our response can alleviate your concerns. Please let us know if you have any additional questions.**

---

> ### Comment · Reviewer_pPDQ · 2024-08-12
> **Thanks for the rebuttal**
>
> Thanks for the author response. I have some quesions:
>
> **Q1** I'm confused about the claim:  "The authors found that directly using pLMs to revise the predicted sequences can only yield a marginal enhancement." Could you give me some evidence to support this?
>
> **Q2** Your goal is to design sequence, therefore, pLM actually brings the potential of data leackage. Could you provide the ablation results by removing the pLM on the CATH dataset? You can use the same model architecture, but do not load pretrained weights.
>
> **Q3** For the new results, ProteinMPNN probably to be weak. Could you provide more baseline comparisons?

---

> ### Author Response · Authors · 2024-08-12
>
> We sincerely appreciate the reviewer for engaging in the discussion and providing further insightful comments which greatly help us to improve our work!
>
> > I'm confused about the claim: "The authors found that directly using pLMs to revise the predicted sequences can only yield a marginal enhancement." Could you give me some evidence to support this?
>
> We apologize for the oversight in accurately citing the source. In the proof-of-concept for LM-Design (Section 2.3, https://proceedings.mlr.press/v202/zheng23a/zheng23a.pdf), the authors studied whether a pre-trained pLM (ESM-1b 650M) is capable of refining predicted sequences generated by the advanced inverse folding model ProteinMPNN. As illustrated in Figure 1C, this approach yields immediate but marginal gains in sequence recovery, increasing from approximately 49% to 50%. While this experiment demonstrates the potential for further leveraging pLMs to enhance protein sequence design, it also indicates that even for the structure of a native protein sequence present in the ESM's pretraining sequence data, ESM still cannot significantly improve the prediction.
>
> > Your goal is to design sequence, therefore, pLM actually brings the potential of data leackage. Could you provide the ablation results by removing the pLM on the CATH dataset? You can use the same model architecture, but do not load pretrained weights.
>
> Thanks for your constructive suggestion. We have performed ablation studies on a non-pretrained model. As it is challenging to train a large model from scratch on limited training data, we opted for the smallest model (i.e., ESM-2 8M), as our backbone network. We quote the results of baselines from Figure 3 of our paper. We found that Bridge-IF, when not pre-trianed, slightly surpasses the performance of the Bridge-IF that underwent pretraining. Additionally, we observed an interesting phenomenon where LM-Design (ESM-2 8M) showed a decrease in performance relative to the ProteinMPNN. These observations suggest that the potential for data leakage is unlikely to contribute to Bridge-IF's superior performance.
>
> |  | recovery rate |
> |-|-|
> | ProteinMPNN | 48.18 |
> | LM-Design (ESM-2 8M) | 47.58 |
> | Bridge-IF (ESM-2 8M) | 51.18 |
> | Bridge-IF (ESM-2 8M w/o pretraining) | **51.27** |
>
> > For the new results, ProteinMPNN probably to be weak. Could you provide more baseline comparisons?
>
> We would like to emphasize that the preliminary results presented in our initial rebuttal aimed to demonstrate that Bridge-IF can also generalize to de novo backbone structures for which the corresponding native sequences are unavailable. This capability addresses the potential concern regarding model memorization of the pre-training sequence data. Besides, it is important to note that de novo protein design presents a considerable challenge, since the de novo structures are not as high-resulution as experimentally determined structures used for model training. Most of the baselines compared in the main paper have not been specifically optimized for this task. Indeed, ProteinMPNN is a very robust model, which is widely utilized in de novo protein design and the evaluation of protein backbone generation models [1,2,3].
>
> Fortunately, during our discussions with Reviewer r7sf, we have dived into the application of the inverse folding model for de novo protein design. We summarize our findings as follows: 1) Adding noise to the training data is crucial for robust de novo protein design; 2) There is a trade-off between diversity and sample quality, with models using a lower sampling temperature achieving better performance on this challenging task. Inspired by these findings, we successfully proposed an improved and robust Bridge-IF model for de novo protein design. Specifically, we trained Bridge-IF with 0.2-Å noise and sampled from it at a lower temperature.
>
> In experiments with structures generated by FrameDiff [2], compared with the state-of-the-art ChromaDesign [4], recently published in Nature, which made a lot of improvements in de novo protein sequences, our Bridge-IF demonstrates comparable performance in terms of scTM and designability (scTM > 0.5), while outperforming on the more stringent designability metric (scRMSD < 2Å).
>
> |  | scTM | designability (scTM > 0.5) | designability (scRMSD < 2Å)
> | - | - | - | - |
> | ChromaDesign  | 0.86 | 0.99 | 0.41 |
> | Bridge-IF | 0.86 | 0.99 | **0.44** |
>
> [1] Watson, Joseph L., et al. "De novo design of protein structure and function with RFdiffusion." Nature 2023.
>
> [2] Yim, Jason, et al. "SE (3) diffusion model with application to protein backbone generation." ICML 2023.
>
> [3] Wu, Kevin E., et al. "Protein structure generation via folding diffusion." Nature communications 2024.
>
> [4] Ingraham, John B., et al. "Illuminating protein space with a programmable generative model." Nature 2023.
>
> **If our response resolves your concerns, we kindly ask you to consider adjusting the scores. Please let us know if you have any additional questions.**

---

> > ### Author Response · Authors · 2024-08-13
> > **Looking forward to your feedback**
> >
> > Dear reviewer,
> >
> > We would like to first express our sincere gratitude for your time and effort in reviewing our paper. We appreciate your valuable suggestions and will incorporate the discussions and results into the final version.
> >
> > Considering the author-reviewer discussion is ending very soon, could you kindly check our responses and let us know if you have further concerns? We are more than willing to address any other concerns or questions.
> >
> > We would greatly appreciate it if the reviewer would consider adjusting the score, based on our response and other review comments.
> >
> > Sincerely, Authors

---

> > ### Comment · Reviewer_pPDQ · 2024-08-13
> > **Thanks for your reply**
> >
> > Thanks for the author rebuttal. I update my comments:
> >
> > **Q1** I can not agree with your claim, that is "pLMs to revise the predicted sequences can only yield a marginal enhancement". You provide me the evidence of  LMDesign, where sequence predictions of the ProteinMPNN were fed as input into the ESM-1b, without any training. In this case, LMDesign improves the recovery from approximately 49% to 50%. **However, with fintuning pLM, LM design increase the performance to 54.62%.** Your model falls into the fintuning case, therefore, your claim is not correct.
> >
> > **Q2** ESM-2 8M may be not enough, you should use the ESM-650M to see how much performance gain is obtained from pLM's pretraining.  In my experience, it is very strange that pLMs that are not pre-trained outperform pre-trained pLMs in this task. I’m not trying to make things difficult for you; I just want you to clarify how much each of pLM and diffusion contributes to the overall improvement. Finetuning ESM-650M is not expensive on CATH dataset.
> >
> > **Q3** ChromaDesign is not targeted to inverse folding. You can try other strong inverse folding baselines on the de-novo dataset. But the rebuttal ddl is approaching, you do not need to do such experiments.

---

> ### Author Response · Authors · 2024-08-13
>
> We appreciate the reviewer for providing insightful comments!
>
> > I can not agree with your claim, that is "pLMs to revise the predicted sequences can only yield a marginal enhancement". You provide me the evidence of LMDesign, where sequence predictions of the ProteinMPNN were fed as input into the ESM-1b, without any training. In this case, LMDesign improves the recovery from approximately 49% to 50%. However, with fintuning pLM, LMDesign increase the performance to 54.62%. Your model falls into the fintuning case, therefore, your claim is not correct.
>
> Thanks for the comment. Due to the intermittent discussion, misunderstandings may have arisen regarding this claim. As stated in our initial rebuttal, our claim aims to indicate that pLMs pre-trained exclusively on sequence data cannot be directly applied to inverse folding problems without fine-tuning, even if they have encountered native sequences during the pre-training phase. Our second rebuttal is conducted to explain in detail.
>
> We believe that this proof-of-concept experiment without fine-tuning can mitigate concerns about the potential for data leakage. If data leakage is significant, the pLMs should demonstrate a substantial improvement in predictions without fine-tuning, as nearly half (49%) of the input sequences were identical to the native sequences. For an analysis of the experiments in the fine-tuning case, please refer to the response below.
>
> > ESM2 8M may be not enough, you should use the ESM-650M to see how much performance gain is obtained from pLM's pretraining. I want you to clarify how much each of pLM and diffusion contributes to the overall improvement.
>
> The reason we chose ESM-2 8M is that the CATH v4.2 dataset only consists of 18,024 proteins for training, and we cannot train a larger Transformer model from scratch. Following your request, we utilized the ESM-2 650M as the backbone and trained it from scratch. However, as expected, Bridge-IF did not work and even produced worse predictions than the prior (50% for PiFold). We wish to emphasize that our performance achieved with pre-trained ESM-2 650M is not the result of information leakage; rather, training large Transformers from scratch presents inherent challenges. Conversely, experiments with ESM-2 8M reveal that Bridge-IF (w/o pre-training) achieved comparable or slightly superior performance compared to Bridge-IF (w/ pre-training). This phenomenon may be due to the inconsistency in the data distribution and training objectives during the pre-training and fine-tuning stages. This finding indicates that the observed performance improvements cannot be attributed to information leakage, thereby validating the efficacy of our model.
>
> |backbone| recovery rate |
> |-|-|
> |ESM2 8M w/o pretraining | **51.27** |
> |ESM2 8M | 51.18 |
> |ESM2 650M w/o pretraining| 46.41|
> |ESM2 650M|**58.59**|
>
> Additionally, we conducted further ablation studies to verify the contribution of our diffusion bridge approach. Bridge-IF significantly outperforms both LM-Design and the traditional diffusion model GraDe-IF while **utilizing the same pLM backbone (ESM-1b 650M)**. This highlights the effectiveness of the Markov bridge in modeling the refinement process. The result for LM-Design is quoted from the original paper, while GraDe-IF was implemented by replacing the denoising network with the same pLM used in Bridge-IF during the early rebuttal stage.
>
> ||Sequence recovery↑|
> |-|-|
> |LM-Design (PiFold + ESM-1b 650M) | 55.65 |
> |GraDe-IF (ESM-1b 650M)|55.37|
> |Bridge-IF (PiFold + ESM-1b 650M)|**58.49**|
>
> > ChromaDesign is not targeted to inverse folding. You can try other strong inverse folding baselines.
>
> We apologize for not providing a clear description of ChromaDesign. Indeed, **ChromaDesign is an inverse folding model**. Chroma is a generative model capable of directly sampling novel protein structures and sequences. It comprises a backbone network trained as a diffusion model to generate structures and a design network (i.e., ChromaDesign, G.3 in https://static-content.springer.com/esm/art%3A10.1038%2Fs41586-023-06728-8/MediaObjects/41586_2023_6728_MOESM1_ESM.pdf) that facilitates robust sequence prediction based on the generated structures (I.2, J.5, and J.6). Both ProteinMPNN and ChromaDesign have demonstrated their ability to generate de novo proteins through wet-lab experiments, showcasing they are strong inverse folding baselines on the de novo protein design. The performance of Bridge-IF surpasses that of ProteinMPNN and ChromaDesign, indicating that it can generalize to de novo proteins, alleviating concerns about information leakage.
>
> We have made every effort to address your concerns regarding data leakage through a proof-of-concept experiment without fine-tuning, ablation studies in the fine-tuning context, and de novo protein design experiments. **If our response resolves your concerns, we kindly ask you to consider adjusting the scores. Please let us know if you have any additional questions**.

---

> > ### Comment · Reviewer_pPDQ · 2024-08-14
> > **Reply to authors**
> >
> > Thanks for the author rebuttal.
> >
> > Your initial rebuttal is "As pLMs such as ESMs are trained solely on sequence data, they cannot directly be applied for the inverse folding problem on their own, regardless of whether or not the proteins are part of a pLM's training data. As shown in the preliminary study of LM-Design, the authors found that directly using pLMs to revise the predicted sequences can only yield a marginal enhancement. Hence, the focus of the recent research is on how to better utilize pLMs to facilitate the inverse folding task." But my question is about data leackage. Your initial rebuttal could not address my concern. You did many things unrelated to the key question, but have not yet convinced me. The best way to evaluate this is to compare an un-pretrained model with a pretrained model.
> >
> > In my request, you provided the results, where ESM2 650M w/o pretraining works even worse than the baseline. But, with pretraining, the model can outperform baseline significantly. This shows the pretraining knowledge performs a key role in improving the performance, this is exactly what I say data leackage. You also use ESM2 8M to prove pretraining do not improve the performance, even resuling in negative affect. However, this is not correlated to your main results, and all related works using ESM-650M find that pretraining is important to the performance.
> >
> > Up to now, I'm still confused how much performace gain comes from diffusion and pretraining, respectively. Therefore, we don't know how the data leakage issue affected your model. In addition, in your rebuttal, LM-Design (ESM-2 8M) even performs worse than ProteinMPNN baseline, which is strange. This either indicates ESM-8M do not suitable for this task, or you have made some wrong in the experiments.
> >
> > Considering above concerns, I retain my score.

---

> ### Author Response · Authors · 2024-08-14
>
> > You did many things unrelated to the key question, but have not yet convinced me. The best way to evaluate this is to compare an un-pretrained model with a pretrained model.
>
> We agree that comparing a non-pretrained model with a pretrained model is a suitable approach (see the below response to other comments for a detailed discussion). However, in our initial rebuttal, we have conducted experiments on **de novo protein design, where the risk of data leakage is nonexistent, as the ESM model has not encountered these novel proteins during pre-training**. Our Bridge-IF outperforms robust and strong baselines ProteinMPNN and ChromaDesign, which have been validated with wet-lab experiments, also providing strong evidence to alleviate concerns regarding data leakage.
>
> > ESM2 650M w/o pretraining works even worse than the baseline. But, with pretraining, the model can outperform baseline significantly. This shows the pretraining knowledge performs a key role in improving the performance, this is exactly what I say data leackage.
>
> The phenomenon that ESM2 650M w/ pretraining significantly outperforms ESM2 650M w/o pretraining can **only demonstrate the importance of pre-training, and it is not evidence for information leakage**. As previously emphasized, **we have only 20k training samples, which is extremely insufficient for training a large Transformer model with 650M parameters from scratch**. In practice, larger Transformers typically require more large-scale datasets for training to achieve good convergence and generalization capabilities [1,2,3]. Due to the lack of inductive biases in Transformers, directly training these large models on small-sized datasets always results in poor performance [2,3,4]. Taking ESM [4] used in our experiment as an example, in its original paper, we observe that **non-pretrained ESM achieved the worst performance across all tasks**, i.e., a performance drop of about 30% compared to the pre-trained version (**Table 2, 3, 4, and 5 in https://www.pnas.org/doi/pdf/10.1073/pnas.2016239118**), which is aligned with our results of Bridge-IF (ESM-2 650M w/o pretraining).
>
> [1] Kaplan, et al. "Scaling laws for neural language models."
>
> [2] Radford, et al. "Improving Language Understanding by Generative Pre-Training."
>
> [3] Dosovitskiy, et al. "An image is worth 16x16 words: Transformers for image recognition at scale."
>
> [4] Rives, et al. "Biological structure and function emerge from scaling unsupervised learning to 250 million protein sequences."
>
> > You also use ESM2 8M to prove pretraining do not improve the performance, even resulting in negative affect. However, this is not correlated to your main results, and all related works using ESM-650M find that pretraining is important to the performance. In your rebuttal, LM-Design (ESM-2 8M) even performs worse than ProteinMPNN, which is strange.
>
> Based on the analysis above, we compare the pretrained and non-pretrained ESM2 8M to **address your concern about data leakage, rather than to prove that pre-training does not enhance performance**. As we only have 20k proteins for training, **when trained from scratch, the smallest ESM2 8M is the only model that achieves good performance**. The comparable performance of pretrained ESM2 8M and non-pretrained ESM2 8M proves that the performance improvement does not stem from the potential for data leakage
>
> The results of LM-Design (ESM-2 8M) and ProteinMPNN are quoted from the original paper of LM-Design (Figure 6, https://proceedings.mlr.press/v202/zheng23a/zheng23a.pdf). The reason for this phenomenon is that even though ESM-2 8M has been pre-trained, **its limited model capacity leads to weaker generalization performance in fine-tuning tasks**. This experiment demonstrates that the improvement achieved through pre-training stems from **scaling capability of large models**, rather than from information leakage.
>
> > I'm still confused how much performance gain comes from diffusion and pretraining, respectively. Therefore, we don't know how the data leakage issue affected your model.
>
> In the previous response, we compared Bridge-IF to LM-Design and GraDe-IF using the **same ESM-1b 650M** to verify the performance gain from our diffusion bridge model. In Figure 3 of our paper, we have shown **consistent performance improvement across pre-trained pLMs with different sizes**, which is acknowledged in your initial review.
>
> In summary, the experimental results indicate that: **1) both pre-training and the diffusion bridge model significantly enhance model performance and 2) the improvement derived from pre-training is primarily attributed to the scaling capability of large models. As the model size increases, it exhibits a greater ability to learn general evolutionary knowledge about proteins from extensive datasets**.
>
> **We hope that our clearer explanation can alleviate your concerns. We kindly ask you to consider adjusting the scores. Please let us know if you have any additional questions.**

---

> > ### Author Response · Authors · 2024-08-14
> >
> > Dear reviewer,
> >
> > We would like to first express our sincere gratitude for your time and effort in reviewing our paper. We appreciate your valuable suggestions and will incorporate the discussions and results into the final version.
> >
> > Considering the author-reviewer discussion is ending very soon, could you kindly check our responses.
> >
> > We would greatly appreciate it if the reviewer would consider adjusting the score, based on our response and other review comments.
> >
> > Sincerely, Authors

---

### Official Review · Reviewer_3QQQ · 2024-07-12

**Soundness:** 3
**Presentation:** 2
**Contribution:** 3
**Rating:** 6
**Confidence:** 4

**Summary:**

In this paper the authors introduce Bridge-IF, an inverse folding approach based on a novel Markov bridge sampling formulation and both protein sequence and structure encoders. They show that leveraging both the information from pre-trained protein language models and a probabilistic, iterative sampling procedure leads to improved perplexity and sequence recovery on the CATH benchmark.

**Strengths:**

Leveraging pre-trained pLMs for IF is a clear and seemingly effective idea. The demonstration in Fig 3 is also one of the clearest results to date on the effectiveness of pLM model size scaling for a downstream task. This is an exciting result and worth highlighting.

**Weaknesses:**

Because of the effectiveness of the pLM in approximating the Markov bridge process, it is not as clear how central the Markov bridge approach itself is to the results, versus augmenting an “autoregressive diffusion” model like ProteinMPNN or a more traditional diffusion model like GraDe-IF. It’s possible that the comparison to LM-design can clarify this point, if the authors include additional discussion. Figures 1 and 2 are particularly helpful for understanding the approach and architecture.

**Questions:**

Can the authors comment on any attempts to apply pLM and structure conditioning to alternative sampling methods, and the relative advantages of adapting the Markov bridge approach?

I’m not sure what the authors mean by “discriminative formulation.” Autoregressive models are still generative models, not “discriminators” in the usual usage referring to classifiers.

**Limitations:**

Yes

---

> ### Author Rebuttal · Authors · 2024-08-06
>
> We deeply appreciate the reviewer for the insightful and constructive comments!
>
> > **W1**. Because of the effectiveness of the pLM in approximating the Markov bridge process, it is not as clear how central the Markov bridge approach itself is to the results, versus augmenting an “autoregressive diffusion” model like ProteinMPNN or a more traditional diffusion model like GraDe-IF. It’s possible that the comparison to LM-design can clarify this point, if the authors include additional discussion.
> >
> > **Q1**. Can the authors comment on any attempts to apply pLM and structure conditioning to alternative sampling methods, and the relative advantages of adapting the Markov bridge approach?
>
> Thanks for pointing this out. To the best of our knowledge, LM-Design is among the first attempts to steer pLMs to perform protein inverse folding. The authors of LM-Design additionally introduced an autoregressive variant by adapting ProGen2, a widely-recognized autoregressive pLM, to function as a structure-based sequence designer (Appendix A.4, https://proceedings.mlr.press/v202/zheng23a/zheng23a.pdf). Their experimental results indicated that LM-Design (ESM-1b) significantly outperforms its autoregressive counterpart, LM-Design (ProGen2), thereby demonstrating that iterative refinement is more suitable for protein modeling than autoregressive methods. Furthermore, our Bridge-IF approach substantially surpasses LM-Design while utilizing the same pLM backbone, highlighting the efficacy of the Markov bridge in modeling the refinement process. Additionally, we implemented a pLM-augmented GraDe-IF by substituting the denoising network (i.e., an equivariant graph neural network) with the same pLM utilized in Bridge-IF. Despite this modification, our Bridge-IF continues to outperform GraDe-IF, underscoring the advantages of Markov bridges over diffusion models in better leveraging structural priors. These results emphasize the superiority of the Markov bridge method compared to other sampling techniques.
>
> || Sequence recovery↑|
> |-|-|
> |LM-Design (ProGen2)|50.80|
> |GraDe-IF (ESM-1b 650M) |55.37|
> |LM-Design (ESM-1b 650M)|55.65|
> |Bridge-IF (ESM-1b 650M)|**58.49**|
>
> > **Q2**. I’m not sure what the authors mean by “discriminative formulation.” Autoregressive models are still generative models, not “discriminators” in the usual usage referring to classifiers.
>
> We apologize for our unclear definition. We adopt the term "discriminative formulation" from ill-posed problems like retrosynthesis and super-resolution, where multiple valid and compatible predictions exist for a single input. In the context of protein inverse folding, we use "discriminative formulation" to emphasize that autoregressive and one-shot models formulate the problem as the prediction of the most probable sequence (typically the native sequence observed in the training data) for a given protein structure.
>
> **We hope our response can alleviate your concerns. Please let us know if you have any additional questions.**

---

> > ### Comment · Reviewer_3QQQ · 2024-08-12
> >
> > I thank the authors for their response and the above clarifications. I have read the response and those to the other reviewers and will keep my score the same.

---

> > > ### Author Response · Authors · 2024-08-14
> > >
> > > We sincerely appreciate your valuable comments which greatly help us to improve our work and will incorporate the discussions and results into the final version.

---

### Author Rebuttal · Authors · 2024-08-07

Dear Area Chairs and Reviewers,

In the author response period, we made diligent efforts to address reviewers' concerns and provided additional experimental results to further verify our contributions. The summary of our main efforts is presented as follows:

- We have provided a detailed explanation for the protocol, experimental details, and implementation details. (To all reviewers)

- We have further conducted diverse experiments (de novo protein design and zero-shot mutation effect prediction) to demonstrate the general applicability of the proposed method (To r7sf, HBUp, pPDQ).

- We have further conducted experiments to analyze the diversity and inference speed to demonstrate the advantages of the proposed method (To r7sf).

- We have further discussed and conducted experiments to analyze the impact of Markov bridge models (To 3QQQ).

- We have clarified the results of baselines (To pPDQ).

- We have expressed our thoughts about potential data leakage in the use of pre-trained protein language models (pLMs), and conducted experiments to demonstrate that performance improvements occur regardless of whether the protein sequences are included in the pLM's training data (To pPDQ, r7sf).

In our individual responses, we provide detailed answers to all the specific questions raised by the reviewers. Further discussions are welcomed to facilitate the reviewing process toward a comprehensive evaluation of our work.

---

### Decision · Program_Chairs · 2024-09-25

**Decision:**

Accept (poster)

**Comment:**

The authors' method seems to work quite well on an important problem, and is applying a technique that--while relatively newly studied in the literature--hasn't been applied to inverse folding before. Ultimately, there's straightforward justification for acceptance: they evaluate on the right datasets, and beat very competitive recent methods like PiFold. Finally, I do generally agree that the authors' results on de novo proteins do demonstrate that their method is quite competitive and generally answers the concern about information leakage. I *do* agree with Reviewer pPDQ that the authors' should include the remaining baselines here in the final version, but am nevertheless generally in favor of acceptance.